# Rethinking Decoders for Transformer-based Semantic Segmentation: A Compression Perspective

**Qishuai Wen and Chun-Guang Li**[*]
School of Artificial Intelligence
Beijing University of Posts and Telecommunications, Beijing 100876, P.R. China
{wqs,lichunguang}@bupt.edu.cn

## Abstract

State-of-the-art methods for Transformer-based semantic segmentation typically adopt Transformer decoders that are used to extract additional embeddings from image embeddings via cross-attention, refine either or both types of embeddings via self-attention, and project image embeddings onto the additional embeddings via dot-product. Despite their remarkable success, these empirical designs still lack theoretical justifications or interpretations, thus hindering potentially principled improvements. In this paper, we argue that there are fundamental connections between semantic segmentation and compression, especially between the Transformer decoders and Principal Component Analysis (PCA). From such a perspective, we derive a white-box, fully attentional DEcoder for PrIncipled semantiC segemenTation (DEPICT), with the interpretations as follows: 1) the self-attention operator refines image embeddings to construct an ideal principal subspace that aligns with the supervision and retains most information; 2) the cross-attention operator seeks to find a low-rank approximation of the refined image embeddings, which is expected to be a set of orthonormal bases of the principal subspace and corresponds to the predefined classes; 3) the dot-product operation yields compact representation for image embeddings as segmentation masks. Experiments conducted on dataset ADE20K find that DEPICT consistently outperforms its black-box counterpart, Segmenter, and it is light weight and more robust. Our code and models are available at https://github.com/QishuaiWen/DEPICT.

## 1 Introduction

Semantic segmentation has been a fundamental task in computer vision for decades. In the supervised setting, the task aims to segment an image into regions corresponding to different predefined classes. The dominant approaches for semantic segmentation have experienced significant shifts, in particular, from hand-crafted features [16] to deep learning, from Convolutional Neural Networks (CNNs) [24, 5] to Vision Transformers (ViT) [46, 35], and then from per-pixel classification to mask classification [8]. Recently, state-of-the-art methods [35, 8, 7, 45] for Transformer-based semantic segmentation [22] typically adopted the Transformer decoders inspired by DETR [2].

Although they vary among different methods, the Transformer decoders typically consist of cross-attention operators that extract additional embeddings (known as class embeddings [35, 45] or mask embeddings [8, 7]) from image embeddings, self-attention operators that refine either or both the additional embeddings and image embeddings, layer normalization (LN) [21] and feedforward neural networks (FFN), which are the default compositions of a Transformer block [37], and one (or two for mask classification) dot-product operation of the two types of embeddings. Here, we illustrate in Figure 1 two representative methods, i.e., Segmenter [35] and MaskFormer [8].

38th Conference on Neural Information Processing Systems (NeurIPS 2024).

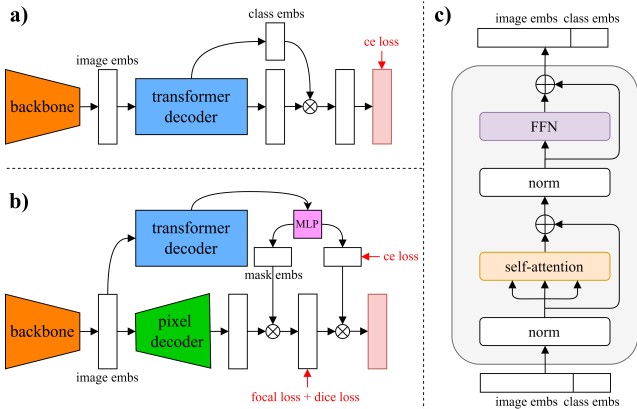

**Figure 1: Illustration for Segmenter and MaskFormer.** a) Segmenter. b) MaskFormer. c) Transformer block adopted by Segmenter. We omit the details of the Transformer decoder adopted by MaskFormer, which refines image embeddings and mask embeddings via self-attention respectively before the cross-attention operations.

Despite the empirical designs of the Transformer decoders being intuitive and having achieved remarkable success [7, 20], there still lack theoretical justifications or interpretations, thus hindering potentially principled improvement (such as identifying and addressing performance bottlenecks). We believe that the first step toward white-box models for Transformer-based semantic segmentation is to answer the following questions: 1) Why do the Transformer decoders outperform a position-wise MultiLayer Perceptron (MLP) that independently classifies each image embedding [35]? 2) What is the underlying mechanism of the self- and cross-attention operators adopted by the Transformer decoders? 3) More importantly, is there a principle for designing and improving the Transformer decoders?

In this paper, we argue that there are fundamental connections between semantic segmentation and compression, especially between the Transformer decoders and Principal Component Analysis (PCA), and that the principle of compression is all we need to derive a white-box decoder akin to the Transformer decoder. To be specific, we extend the objectives of PCA in the geometric and rank minimization views [38] to the context of the coding rate [11, 25, 43] and, following the derivation in CRATE [42], we derive self- and cross-attention operators for semantic segmentation by unrolling the optimization of these objectives.

**Contributions.** The contributions of the paper are highlighted as follows.

1. We take a further step along the fundamental connections between semantic segmentation and compression, by introducing PCA for understanding the empirical designs of decoders for Transformer-based semantic segmentation.
2. We extend the objectives as well as the idea of PCA in terms of the coding rate to unrolled optimization, and thus derive a family of white-box fully attentional DEcoder for PrIncipled semantiC segmenTation (DEPICT).
3. We conduct extensive experiments to evaluate the performance of our DEPICT and find that our DEPICT consistently outperforms its black-box counterpart and shows desirable properties suggested by the derivation.

## 2   Related Work

### 2.1   Interpretability of Decoders for Semantic Segmentation

We cast decoders for semantic segmentation into four categories: 1) convolutional decoders [24, 33, 5, 46]; 2) MLP-based decoders [35, 39, 6, 26]; 3) Transformer decoders [35, 8, 7, 45]; 4) clustering decoders [49, 41, 50, 23, 17]. The core operations of convolutional decoders are convolution and pooling. Despite rooting in signal processing, the interpretability of the learned convolutional filters is limited. Among these decoders, segmentation masks are also features. And, as a self-attention layer can express any convolutional layer [9], the self-attention operator we derive is very likely to be decomposed into convolutional filters. Due its powerful fitting ability, an MLP-based

decoder is quite difficult to interpret. Nonetheless, a single linear layer which is used by pre-trained models [15, 18, 3, 28] as the classifier can be viewed as a parametric softmax projection, which learns a prototype for each class [49].

To our knowledge, Segmenter [35] proposed the first Transformer decoder, which is used to perform per-patch classification; whereas MaskFormer [8] formulated the task of mask classification, which performs clustering at first and then classifies the clusters. As a much broader concept, clustering decoders include all methods that are explicitly based on the idea of clustering. In particular, [41] reformulates cross attention as a clustering process; whereas [17] exploits the principal directions for segmentation, based on the relationship between PCA and $k$-means [13]. In this paper, we instead take a compression perspective, which is more fundamental than clustering.

## 2.2 White-Box Models based on Coding Rate

The coding rate [11] is an effective criterion for compression and is first introduced for segmentation by [25]. And a solid justification for the connections between image segmentation and compression can be traced back to [32], which argues that the optimal segmentation of an image is the one that will give the shortest coding length for encoding the image. Then, a principled framework, termed Maximal Coding Rate Reduction (MCR$^2$) [43], was proposed to learn discriminative and diverse representations, in which MCR$^2$ maximizes the difference between the coding rate of the ambient space and the sum of the coding rate for each class-specific individual subspace and is shown to promote representations to lie in a union of orthogonal subspaces.

By unrolling the gradient-based optimization procedure of MCR$^2$, a deep architecture, termed ReduNet [4] is derived as a white-box counterpart to the black-box of ResNets and CNNs, followed by CRATE [42] for ViTs [15] and CRATE-MAE [29] for masked autoencoders [18]. Intriguingly, CRATE shows a segmentation emergence similar to DINO [3, 44], which is mainly attributed to the Multi-head Subspace Self-Attention (MSSA) operator derived by [42]. These works have laid a solid foundation for further investigation of the connections among compression, representation learning, and vision tasks at varying granularity. Especially, the role of normalization, e.g., ensuring the effectiveness of the coding rate, has been discussed in [1, 4]. As an analog, we adopt LayerNorm to normalize the scale of image embeddings before all attention operators that we derive.

# 3 Our Methods

## 3.1 Notations and Preliminaries

Given an arbitrary image for semantic segmentation, we use $\boldsymbol{Z} = [\boldsymbol{z}_1, \ldots, \boldsymbol{z}_N] \in \mathbb{R}^{D \times N}$ to denote a set of image embeddings, where each image embedding $\boldsymbol{z}_i \in \mathbb{R}^D$ represents one of the $N$ regular non-overlapping patches to which the image is split. Specifically, we assume that $\boldsymbol{Z}$ is zero mean, i.e., $\boldsymbol{Z}\mathbf{1} = \mathbf{0}$, where $\mathbf{1} \in \mathbb{R}^N$ is the vector of 1's. For a Transformer decoder, we use $\boldsymbol{Z}_0$ to denote the input, which is actually the output of the ViT backbone, and $\boldsymbol{Z}_\ell$ is for $\boldsymbol{Z}_0$ after being updated $\ell$ times, where $\ell \leq L_1$. We use $\boldsymbol{Q} = [\boldsymbol{q}_1, \ldots, \boldsymbol{q}_K] \in \mathbb{R}^{D \times K}$ to denote the additional embeddings (or queries, or more precisely, cluster centroids), $\boldsymbol{Q}_0$ for their initialization, $\boldsymbol{Q}_\ell$ for $\boldsymbol{Q}_0$ after being updated $\ell$ times, where $\ell \leq L_2$, and $\boldsymbol{I}$ for an identify matrix with proper dimension. We specially set $K$ equal to the number of predefined classes, $C$, thus referring to the additional embeddings as class embeddings, or classifiers instead. A generalized case will be discussed in Appendix A.3.

For PCA, what we concern about are the leading $C$ principal directions of $\boldsymbol{Z}$ and the associated $C$-dimensional principal subspace, which is denoted as $\mathcal{S}$. For convenience, we simply refer to them as the principal directions and the principal subspace. And we use $\boldsymbol{U} = [\boldsymbol{u}_1, \ldots, \boldsymbol{u}_C] \in \mathbb{R}^{D \times C}$ to denote an arbitrary set of orthonormal bases of $\mathcal{S}$. Notably, we introduce PCA from two different perspectives here and will extend them to the context of coding rate in the following sections. From a geometric perspective, PCA minimizes the squared reconstruction error when recovering $\boldsymbol{Z}$, i.e.,

$$\min_{\boldsymbol{U}, \boldsymbol{Y}} \|\boldsymbol{Z} - \boldsymbol{U}\boldsymbol{Y}\|_F^2, \quad \text{s.t.} \quad \boldsymbol{U}^\top \boldsymbol{U} = \boldsymbol{I}_C. \tag{1}$$

From a rank minimization perspective, it seeks a low-rank approximation of $\boldsymbol{Z}$, i.e.,

$$\min_{\boldsymbol{A}} \|\boldsymbol{Z} - \boldsymbol{A}\|_F^2, \quad \text{s.t.} \quad \text{rank}(\boldsymbol{A}) = C. \tag{2}$$

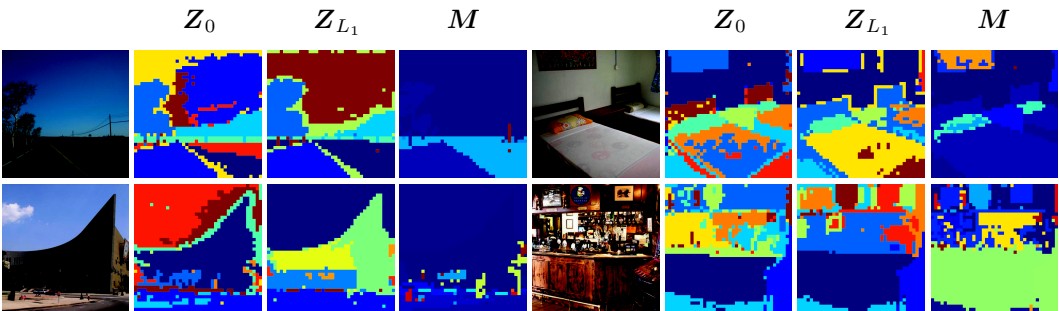

$$\boldsymbol{Z}_0 \qquad\qquad \boldsymbol{Z}_{L_1} \qquad\qquad \boldsymbol{M} \qquad\qquad\qquad \boldsymbol{Z}_0 \qquad\qquad \boldsymbol{Z}_{L_1} \qquad\qquad \boldsymbol{M}$$

**Figure 2: Image Segmentation via PCA and DEPICT.** Given an image, we segment it via PCA and our DEPICT. We perform PCA on its representations $\boldsymbol{Z}_0$ and $\boldsymbol{Z}_{L_1}$, respectively, where the first 10 principal directions are used as cluster centroids. We find that PCA can serve as an effective method for image segmentation especially on the refined features, like $\boldsymbol{Z}_{L_1}$. We also observe that performing PCA on $\boldsymbol{Z}_0$ is more likely to lead to an over-segmentation, which indicates that its principal subspace is not ideal.

Finally, we introduce the concept of the coding rate of $\boldsymbol{Z}$ subject to a certain distortion $\epsilon > 0$, which is calculated as follows:

$$R(\boldsymbol{Z}) \doteq \frac{1}{2}\log\det(\boldsymbol{I}_D + \frac{D}{N\epsilon^2}\boldsymbol{Z}\boldsymbol{Z}^\top). \tag{3}$$

### 3.2 Bridging the Transformer Decoders and PCA

For classification tasks, such as semantic segmentation, on the scale of an entire dataset, it is desirable to have a Linear Discriminative Representation (LDR), which can be well modeled by a union of orthogonal subspaces [43, 4]. However, it is difficult to explicitly model such a desirable structure when $C$ is relatively large, even at the cost of sacrificing the diversity of each class (i.e., the dimensions of each subspace). Fortunately, we notice that a Transformer decoder segments each image independently; that is, an embedding of one image would never attend to or interact with the embeddings from a different image. Additionally, despite that the intra-class variance or diversity is very rich for semantic segmentation on entire dataset, it would be limited within a single image. Therefore, we focus on one single image for now and then generalize it in Section 3.5.

As shown in Figure 1, the Transformer decoders for semantic segmentation typically project image embeddings onto additional embeddings to predict masks. In the case we focus on, it is projected onto the $C$-dimensional subspace spanned by class embeddings, where $C$ is much smaller than $D$. Therefore, we can view semantic segmentation as a process of dimension reduction, i.e., compression, where the masks $\boldsymbol{M} = \boldsymbol{Q}^\top\boldsymbol{Z} \in \mathbb{R}^{C\times N}$ represent more compact features compared to $\boldsymbol{Z}$. Intuitively, the subspace is expected to contain as much information as possible, which is crucial for capturing the rich intra-class variance for semantic segmentation. Meanwhile, it is also desirable for the class embeddings to be orthonormal for more discriminative classification [19, 30, 43, 51]. In other words, we seek to find a low-dimensional subspace that best fits the image embeddings, which is exactly the idea behind PCA, and to find a set of orthonormal bases of it as classifiers.

It is not difficult to show that the problem in (1) is equivalent to:

$$\max_{\boldsymbol{U}} \frac{1}{N}\text{trace}(\boldsymbol{U}^\top\boldsymbol{Z}\boldsymbol{Z}^\top\boldsymbol{U}) := \sum_{c=1}^{C}\text{Var}(\boldsymbol{u}_c^\top\boldsymbol{Z}), \quad \text{s.t.} \quad \boldsymbol{U}^\top\boldsymbol{U} = \boldsymbol{I}_C, \tag{4}$$

which indicates that the principal subspace should retain most variance (i.e., information), and the variances after projection onto the principal directions are maximized. Therefore, we contend that the principal subspace and the subspace spanned by class embeddings, as well as the principal directions and class embeddings, are fundamentally related; in an ideal case, they are equivalent. In Figure 2, we visualize several images segmented using PCA and find that PCA can indeed serve as an effective method, validating the above analysis.

In contrast to the Transformer decoders, a single linear layer can be viewed as performing PCA on the entire dataset, at which scale the image embeddings definitely cannot be well fitted by a $C$-dimensional subspace (i.e., performing a bad compression). Meanwhile, as the intra-class variance in semantic segmentation is richer than in image classification, a single static classifier (or prototype,

or more precisely, basis) is not sufficient to represent a class well. This is why a single linear layer clearly lags behind more complex methods in fine-grained classification tasks.

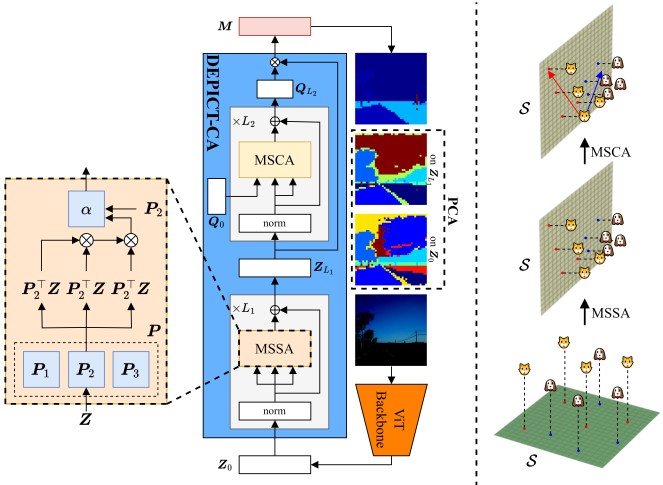

**Figure 3: Illustration for DEPICT.** Given an image for semantic segmentation, we represent it as $\boldsymbol{Z}_0$ by the ViT backbone. Segmenting it by performing PCA on $\boldsymbol{Z}_0$, we find that $\mathcal{S}$ of $\boldsymbol{Z}_0$ is not ideal. We thus adopt the MSSA operator to refine the image embeddings, iteratively constructing an ideal $\mathcal{S}$. Performing PCA again on $\boldsymbol{Z}_{L_1}$, we find that the segmentation results are improved. Then, we adopt the MSCA operator to find a low-rank approximation of $\boldsymbol{Z}_{L_1}$ that lies in $\mathcal{S}$ as classifiers. For example, we use the dogs and cats on the right to represent image embeddings of two different classes in the feature space. Initially, the projections of dogs and cats onto $\mathcal{S}$ are not well linearly separable. DEPICT, however, constructs an ideal $\mathcal{S}$ and effectively classify them.

### 3.3 Constructing an Ideal Principal Subspace via Self-Attention

Despite PCA being an effective method, which is also observed in [17], we find that performing it directly on $\boldsymbol{Z}_0$ typically yields inferior segmentation results, such as oversegmentation, as shown in Figure 2. We attribute the reason for this to an less ideal principal subspace of $\boldsymbol{Z}_0$, which means that the relevant information for supervised semantic segmentation is either not contained in it or not significant enough. This hypothesis is intuitive since $\boldsymbol{Z}_0$ is expected to be generic features and requires refinement to align with specific supervision. To this end, we propose to refine it to construct an ideal principal subspace. As shown in Figure 2, the refinement can in fact alleviate the over-segmentation and improve the segmentation results.

Specifically, we assume that $\boldsymbol{P} = [\boldsymbol{p}_1, \ldots, \boldsymbol{p}_C] \in \mathbb{R}^{D \times C}$ is a set of orthonormal bases of an ideal subspace. Now we optimize the objective of (4) with respect to (w.r.t.) $\boldsymbol{Z}$ to ensure that $\boldsymbol{P}$ is the principal subspace of $\boldsymbol{Z}$ after refinement, that is,

$$\max_{\boldsymbol{Z}} \sum_{c=1}^{C} \mathrm{Var}(\boldsymbol{p}_c^\top \boldsymbol{Z}), \tag{5}$$

where $\boldsymbol{P}$ is learned through backpropagation during training. However, instead of optimizing $\boldsymbol{Z}$ in (5) directly, we choose to maximize the projected coding rate onto these bases, that is,

$$\max_{\boldsymbol{Z}} \sum_{c=1}^{C} R(\boldsymbol{p}_c^\top \boldsymbol{Z}) := \sum_{c=1}^{C} \frac{1}{2} \log \det(1 + \frac{1}{N\epsilon^2} \boldsymbol{p}_c^\top \boldsymbol{Z} \boldsymbol{Z}^\top \boldsymbol{p}_c). \tag{6}$$

This is because the coding rate is a more generalized metric and is effective in high-dimensional spaces. Still, it is equivalent to (5) (as $\boldsymbol{p}_c^\top \boldsymbol{Z} \boldsymbol{Z}^\top \boldsymbol{p}_c$ is a non-negative scalar). According to the seminal work [42], optimizing problem (6) via a gradient step with a learnable step size $\alpha > 0$ derives the MSSA operator, which in our case can be written as:

$$\boldsymbol{Z}^{\ell+1} = (1 + \frac{\alpha}{N\epsilon^2})\boldsymbol{Z}^\ell - \frac{\alpha}{N\epsilon^2} \cdot \mathrm{MSSA}(\boldsymbol{Z}^\ell \mid \boldsymbol{P}), \tag{7}$$

where the MSSA operator is defined as follows:

$$\mathrm{MSSA}(\boldsymbol{Z} \mid \boldsymbol{P}) \doteq \frac{1}{N\epsilon^2} \cdot [\boldsymbol{p}_1, \ldots, \boldsymbol{p}_C] \begin{bmatrix} \mathrm{SSA}(\boldsymbol{Z} \mid \boldsymbol{p}_1) \\ \vdots \\ \mathrm{SSA}(\boldsymbol{Z} \mid \boldsymbol{p}_C) \end{bmatrix}, \tag{8}$$

$$\mathrm{SSA}(\boldsymbol{Z} \mid \boldsymbol{p}_c) \doteq (\boldsymbol{p}_c^\top \boldsymbol{Z}) \cdot \mathrm{softmax}\left((\boldsymbol{p}_c^\top \boldsymbol{Z})^\top (\boldsymbol{p}_c^\top \boldsymbol{Z})\right), \quad \text{for} \quad c \in \{1, \ldots, C\}. \tag{9}$$

In (8), however, MSSA demands $C$ attention heads, each of which calculates an attention matrix in the shape of $N \times N$. As $C$ is always much larger than the number of heads of Multi-Head Self-Attention (MHSA) [37], which is the black-box counterpart of the MSSA operator, thus the computation of (8) occupies an unacceptable amount of GPU memory.

To remedy this issue, we propose to maximize a lower bound of the sum of the projected coding rate onto the bases in groups, rather than maximizing them directly one by one. By our proof in Appendix D.1, the sum of the projected coding rate onto a set of bases can be bounded below by the projected coding rate onto the subspace they span, i.e.,

$$\sum_{m=1}^{M} R(\boldsymbol{p}_m'^\top \boldsymbol{Z}) \geq R(\boldsymbol{P}'^\top \boldsymbol{Z}) - \gamma, \tag{10}$$

where $\boldsymbol{P}' = [\boldsymbol{p}_1', \ldots, \boldsymbol{p}_M'] \in \mathbb{R}^{D \times M}$ consists of $M$ different bases in $\boldsymbol{P}$, and $\gamma$ is a product of $M(M-1)$ and a constant w.r.t. $\boldsymbol{Z}$. We thus divide the columns of $\boldsymbol{P}$ into $H = C/M$ non-overlapping groups, denoted as $\boldsymbol{P}_1', \ldots, \boldsymbol{P}_H'$, and maximize the lower bound for each group, respectively. We thus reformulate the MSSA operator in our case as:

$$\boldsymbol{Z}^{\ell+1} = (1 + \alpha \frac{M}{N\epsilon^2})\boldsymbol{Z}^\ell - \alpha \frac{M}{N\epsilon^2} \cdot \mathrm{MSSA}(\boldsymbol{Z}^\ell \mid \boldsymbol{P}), \tag{11}$$

$$\mathrm{MSSA}(\boldsymbol{Z} \mid \boldsymbol{P}) \doteq \frac{M}{N\epsilon^2} \cdot [\boldsymbol{P}_1', \ldots, \boldsymbol{P}_H'] \begin{bmatrix} \mathrm{SSA}(\boldsymbol{Z} \mid \boldsymbol{P}_1') \\ \vdots \\ \mathrm{SSA}(\boldsymbol{Z} \mid \boldsymbol{P}_H') \end{bmatrix}, \tag{12}$$

$$\mathrm{SSA}(\boldsymbol{Z} \mid \boldsymbol{P}_h') \doteq (\boldsymbol{P}_h'^\top \boldsymbol{Z}) \cdot \mathrm{softmax}\left((\boldsymbol{P}_h'^\top \boldsymbol{Z})^\top (\boldsymbol{P}_h'^\top \boldsymbol{Z})\right), \quad \text{for} \quad h \in \{1, \ldots, H\}. \tag{13}$$

So far, we have derived a self-attention operator (12) which takes a gradient step toward constructing an ideal principal subspace. In Appendix A.2, we discuss the differences among MHSA, the original MSSA, and our modified MSSA in (12). Additionally, our goal can also be achieved by minimizing the projected coding rate onto the bases of the orthogonal complement of the ideal principal subspace, with the only change being to set $\alpha < 0$. Therefore, we do not constrain the sign of $\alpha$ in our implementation.

## 3.4 Finding a Low-Rank Approximation via Cross-Attention

With the ideal principal subspace learned via back-propagation and constructed via self-attention, the remaining problem is to find a set of classifiers (i.e., class embeddings) for semantic segmentation. As discussed in Section 3.2, the principal directions would be a desirable choice. However, as shown in Figure 2, PCA still yields inferior segmentation results compared to parametric and learnable methods. We attribute this issue to the principal directions not being flexible to align with supervision, despite the fact that the constructed principal subspace has been. Meanwhile, being an unsupervised method, PCA requires an additional and challenging step that assigns each principal direction a predefined class. To this end, we derive an operator to extract a set of classifiers that satisfy the requirements as follows: 1) learns to align with supervision; 2) span the principal subspace; 3) effectively abstract or represent the image embeddings.

For the last requirement, we propose to optimize an objective that seeks to find a low-rank approximation of $\boldsymbol{Z}$ in terms of the coding rate, i.e.,

$$\min_{\overline{\boldsymbol{Q}}} |R(\boldsymbol{Z}) - R(\overline{\boldsymbol{Q}})|, \quad \text{s.t.} \quad \mathrm{rank}(\overline{\boldsymbol{Q}}) = C, \tag{14}$$

where $\overline{\boldsymbol{Q}} \in \mathbb{R}^{D \times N}$. This is inspired by the rank minimization perspective of PCA in (2), and the objective of $k$-means which can be written as

$$\min_{\overline{\boldsymbol{V}}} \|\boldsymbol{Z} - \overline{\boldsymbol{V}}\|_F^2 \quad \text{s.t.} \quad \mathrm{rank}(\overline{\boldsymbol{V}}) = C, \tag{15}$$

where the columns of $\overline{\boldsymbol{V}} \in \mathbb{R}^{D \times N}$ consist of cluster centroids. Using the coding rate to measure the approximation, the objective (14) is more generalized than (2) and (15). In Appendix D.2, we prove that both $k$-means cluster centroids and the principal directions are reasonably good (or even the optimal) solutions under certain conditions, for (14).

Intuitively, since that the columns of $\overline{\boldsymbol{Q}}$ span a subspace with which dimension is lower than that of $\boldsymbol{Z}$, we have $R(\boldsymbol{Z}) \geq R(\overline{\boldsymbol{Q}})$, which will be validated by the experiments in the Appendix C. Thus, we equivalently maximize the coding rate of $\overline{\boldsymbol{Q}}$ and derive that:

$$\overline{\boldsymbol{Q}}^{\ell+1} = (1 + \alpha \frac{D}{N\epsilon^2})\overline{\boldsymbol{Q}}^\ell - \alpha(\frac{D}{N\epsilon^2})^2 \overline{\boldsymbol{Q}}^\ell \text{softmax}(\overline{\boldsymbol{Q}}^{\ell\top}\overline{\boldsymbol{Q}}^\ell), \tag{16}$$

which is similar to (12). Note that compared to using the Frobenius norm in (2) and (15), the approximation in (14) in terms of the coding rate is relatively loose due to its invariant property [43], thus optimizing over $\overline{\boldsymbol{Q}}$ turns out to be irrelevant to $\boldsymbol{Z}$. Therefore, we replace some $\overline{\boldsymbol{Q}}$ in (16) with $\boldsymbol{Z}$ to further encourage $\overline{\boldsymbol{Q}}$ to approximate $\boldsymbol{Z}$, and thus we have:

$$\overline{\boldsymbol{Q}}^{\ell+1} = (1 + \alpha \frac{D}{N\epsilon^2})\overline{\boldsymbol{Q}}^\ell - \alpha(\frac{D}{N\epsilon^2})^2 \boldsymbol{Z}^\ell \text{softmax}(\boldsymbol{Z}^{\ell\top}\overline{\boldsymbol{Q}}^\ell). \tag{17}$$

Rather than $\overline{\boldsymbol{Q}}$ which is redundant, what we are indeed concerned with is $\boldsymbol{Q}$. Thus, we simplify (17) by updating each $\boldsymbol{q}_i$ only once, i.e.,

$$\boldsymbol{Q}^{\ell+1} = (1 + \alpha \frac{D}{N\epsilon^2})\boldsymbol{Q}^\ell - \alpha(\frac{D}{N\epsilon^2})^2 \boldsymbol{Z}^\ell \text{softmax}(\boldsymbol{Z}^{\ell\top}\boldsymbol{Q}^\ell). \tag{18}$$

For the first requirement, we set $\boldsymbol{Q}_0$ by learnable parameters; in other words, the alignment to predefined classes is learned by adjusting the starting point of gradient optimization. For the second requirement, we confine the updates of $\boldsymbol{Q}$ to the principal subspace of $\boldsymbol{Z}$ by adding projections onto $\boldsymbol{P}$. Similarly to (12), we reformulate (18) as:

$$\boldsymbol{Q}^{\ell+1} = (1 + \alpha \frac{M}{N\epsilon^2})\boldsymbol{Q}^\ell - \alpha \frac{M}{N\epsilon^2}\text{MSCA}(\boldsymbol{Q}^\ell \mid \boldsymbol{Z}, \boldsymbol{P}), \tag{19}$$

$$\text{MSCA}(\boldsymbol{Q} \mid \boldsymbol{Z}, \boldsymbol{P}) \doteq \frac{M}{N\epsilon^2} \cdot [\boldsymbol{P}'_1, \ldots, \boldsymbol{P}'_H] \begin{bmatrix} \text{SCA}(\boldsymbol{Q} \mid \boldsymbol{Z}, \boldsymbol{P}'_1) \\ \vdots \\ \text{SCA}(\boldsymbol{Q} \mid \boldsymbol{Z}, \boldsymbol{P}'_H) \end{bmatrix}, \tag{20}$$

$$\text{SCA}(\boldsymbol{Q} \mid \boldsymbol{Z}, \boldsymbol{P}'_h) \doteq (\boldsymbol{P}'_h{}^\top \boldsymbol{Z}) \cdot \text{softmax}\left((\boldsymbol{P}'_h{}^\top \boldsymbol{Z})^\top (\boldsymbol{P}'_h{}^\top \boldsymbol{Q})\right), \text{ for } h \in \{1, \ldots, H\}, \tag{21}$$

which we refer to as Multihead Subspace Cross-Attention (MSCA).

## 3.5 Decoder for Principled Semantic Segmentation

From a single image to the entire dataset, we should consider multiple "principal subspaces" [38] and thus allow $\boldsymbol{P}$ to model more bases by raising the number of columns of $\boldsymbol{P}$ from $C$ to a hyperparameter $K$ that is larger than $C$. We expect that the principal subspace, as well as the class embeddings, of different images vary, but the class embeddings of the same class lie in a low-dimensional subspace, and all the subspaces of class are orthogonal, thus satisfying our anticipation for LDR on the entire dataset.

By stacking and combining the two steps of Section 3.3 and 3.4, we have a fully attentional white-box decoder as shown in Figure 3, which iteratively constructs an ideal principal subspace by refining image embeddings via the self-attention operators, and then find a low-rank approximation of the refined image embeddings that lies in the principal subspace and corresponds to predefined classes via the cross-attention operators. As our derivations demonstrate that the principle of compression is all we need for designing the decoders, we refer to our approach as the DEcoder for PrIncipled semantiC segmenTation (DEPICT), and our approach described above that extracts additional embeddings via cross-attention is referred to as DEPICT-CA.

We note that the additional embeddings can also be extracted by self-attention [15, 35, 12]; that is, concatenating the two types of embeddings and updating them simultaneously via self-attention alone, as shown in Figure 1 a). In Appendix D.3, we prove that it implicitly performs cross-attention, thus it can be interpreted by our derivations in Section 3.4. We implement a simpler variant of DEPICT that extracts the additional embeddings via self-attention and refer to it as DEPICT-SA.

# 4 Experiments

## 4.1 Segmentation Performance

**Baselines and implementation details.** To evaluate the segmentation performance of our DEPICT, we conduct extensive experiments on datasets ADE20K [47], Cityscapes [10], and Pascal Context datasets [27] compared to Segmenter and MaskFormer. As shown in Figures 1 and 3, Segmenter serves as the black-box counterpart to DEPICT, while MaskFormer is a more advanced method that adopts a pixel decoder, mask classification formulation, and hierarchical backbones and advanced loss functions. For fair comparisons to Segmenter, we use the same settings, including backbones, data augmentation, optimization and inference. We humbly refer readers to [35] for more details. On ADE20K, we use three MSSA layers for DEPICT-SA and three MSSA layers followed by three MSCA layers for DEPICT-CA on most variants, whereas the exceptions and more settings on the other two datasets can be found in Appendix B.

Table 1: **Comparison on ADE20K validation set.** We compare our DEPICT with Segmenter and MaskFormer. The best-performing result of DEPICT is highlighted.

| Model | Backbone | Resolution | mIoU(ss/ms) | #params | FLOPs |
|---|---|---|---|---|---|
| Segmenter | ViT-T | 512x512 | 38.1/38.8 | 1M | 2.2G |
| | ViT-S | 512x512 | 45.3/46.9 | 4M | 6.7G |
| | ViT-B | 512x512 | 48.5/50.0 | 16M | 22.3G |
| | ViT-L | 640x640 | **51.8/53.6** | 28M | 60.4G |
| MaskFormer | Swin-L | 640x640 | **54.1/55.6** | 15M | 47.7G |
| **DEPICT-CA** | ViT-T | 512x512 | 38.4/39.4 | 0.2M | 1.4G |
| | ViT-S | 512x512 | 45.8/47.0 | 0.6M | 3G |
| | ViT-B | 512x512 | 49.0/50.0 | 1M | 4G |
| | ViT-L | 640x640 | **52.8/53.3** | 2.5M | 16.5G |
| **DEPICT-SA** | ViT-T | 512x512 | 39.3/40.7 | 0.2M | 2.9G |
| | ViT-S | 512x512 | 46.7/47.7 | 0.4M | 3.4G |
| | ViT-B | 512x512 | 49.2/50.7 | 0.8M | 4.2G |
| | ViT-L | 512x512 | 52.5/54.0 | 2M | 9.5G |
| | | 640x640 | **52.9/54.3** | 2M | 17.8G |

Table 2: **Comparison on the validation sets of Cityscapes and Pascal Context.** All are based on ViT-L.

| Model | Cityscapes | | | PascalContext | | |
|---|---|---|---|---|---|---|
| | mIoU(ss/ms) | #params | FLOPs | mIoU(ss/ms) | #params | FLOPs |
| Segmenter | 79.1/81.3 | 15.8M | 45.2G | 58.1/59.0 | 28.4M | 30.0G |
| **DEPICT-SA** | 78.8/81.0 | 0.1M | 2.4G | 57.9/58.6 | 1.6M | 5.6G |

Table 3: **Comparison on a subset of ADE20K training set.** All are based on ViT-S.

| Model | Dataset Size | | | | |
|---|---|---|---|---|---|
| | 4K | 8K | 12K | 16K | 20K |
| Segmenter | 38.31 | 41.87 | 43.42 | 44.61 | 45.37 |
| **DEPICT-SA** | 36.42 | 41.75 | 43.34 | 45.12 | 46.72 |
| **DEPICT-CA** | 35.68 | 41.11 | 42.33 | 44.61 | 45.76 |

**Performance comparisons and analysis.** From Table 1 we read that all variants of DEPICT outperform Segmenter with significantly fewer parameters and FLOPS on ADE20K validation set. In

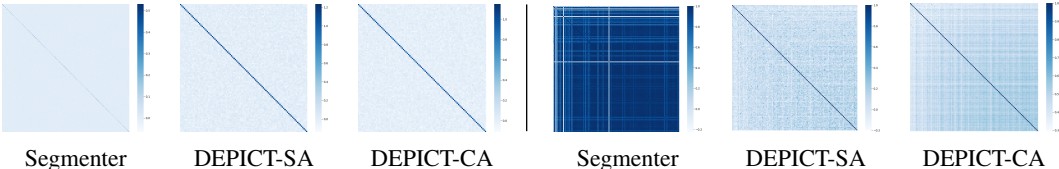

| Segmenter | DEPICT-SA | DEPICT-CA | | Segmenter | DEPICT-SA | DEPICT-CA |

**Figure 4: Investigating orthogonality in DEPICT.** *Left*: $P^\top P$; *Right*: $Q^\top Q$. All variants are based on ViT-L. Since that the MHSA operator contains three parameter matrices, unlike MSSA which has only one, we visualize the matrix responsible for transforming queries. Notably, all the $Q$'s are normalized, whereas $P$ is not.

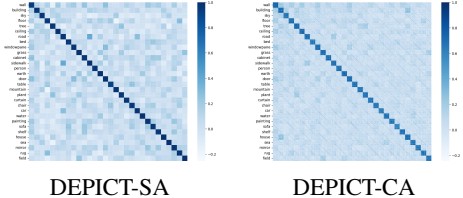

| DEPICT-SA | DEPICT-CA |

**Figure 5: Inner product of class embeddings across images.** We group the class embeddings by their classes and visualize the inner-product among them. We exemplify 30 classes across 100 images. All variants are based on ViT-L.

particular, the best-performing variant, DEPICT-SA based on ViT-Large with an input resolution of 640×640, surpasses Segmenter by 1.1/0.7 mIoU for single/multi-scale inference, while using only 1/14 of the parameters and 1/3 of the FLOPs. We mainly attribute the efficiency of DEPICT to its removal of the FFN block, which plays no role in our interpretations. Such a redundancy has been observed empirically in the field of NLP [31]. but our ablation study in Appendix C shows that naively adopting MSSA while removing the FFN from Segmenter results in worse performance. Moreover, [14] proves that pure attention causes the rank of tokens to decrease rapidly with depth, which is desirable for us to construct an ideal principal subspace. In particular, according to [35], both a finer segmentation granularity and a more performant backbone can significantly boost segmentation performance, partly explaining the inferiority of DEPICT compared to MaskFormer. In Table 2, we find that PICT performs slightly worse than Segmenter. We believe that this is due to the limited size of the dataset; the training set sizes of ADE20K, Cityscapes, and Pascal Context are about 20K, 3K and 5K, respectively. As Table 3 shows, DEPICT requires a sufficient amount of data to demonstrate its superiority.

## 4.2 Desirable Properties of DEPICT

**Orthogonal properties.** According to our interpretations, the parameter matrices of attention are consisted of orthonormal bases, and the extracted class embeddings are fundamentally related to the principal directions, which are also orthonormal. As shown in Figure 4, both $P$ and $Q$ are very close to being orthonormal in DEPICT. Although the parameter vectors learned by Segmenter are also nearly orthogonal, their norms are too small. Despite there are semantic similarities among the predefined classes, the class embeddings of Segmenter are excessively related. Moreover, as we

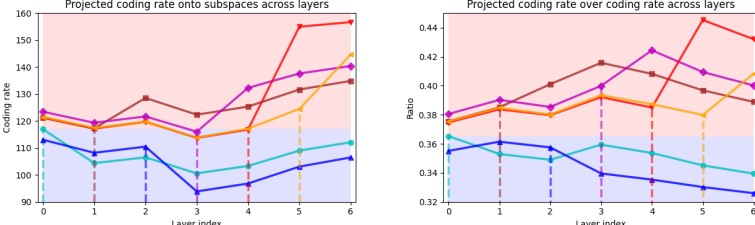

**Figure 6: Measuring the coding rate across layers.** Given the $\ell$-th layer, we use the parameter matrix of its first head, say $P_1'^{\ell}$, and measure the mean projected coding rate of image embeddings onto the subspace spanned by $P_1'^{\ell}$. Each polyline reflects the changes of the projected coding rate onto a particular subspace and the layer index of the subspace is indicated by a vertical dash line in the same color. *Left*: $R(P_1'^{\ell\top} Z)$, across layers. *Right*: $R(P_1'^{\ell} P_1'^{\ell\top} Z)/R(Z)$, across layers.

expected in Section 3.5 and shown in Figure 5, the class embeddings of different classes nearly lie in a union of orthogonal subspaces, and thus the image embeddings are very likely to satisfy assumption for LDR.

**Table 4: Investigating robustness of DEPICT under parameter perturbation.** We experiment with four types of perturbations: 1) the $P$ of each attention operator undergoes a random orthogonal transformation, $O_K \in \mathbb{R}^{K \times K}$; 2) the $P'_h$ of each head undergoes a random orthogonal transformation, $O_M \in \mathbb{R}^{M \times M}$; 3) orthogonalizing $P'_h$; 4) adding random noise from a Gaussian distribution with zero mean and variance $\sigma$ for each parameter independently. The baseline is Segmenter, and we mark the improvements in the table.

| DEPICT-SA | Backbone | | |
|---|---|---|---|
| | ViT-S | ViT-B | ViT-L |
| $P'_h O_M$ | 46.7(+1.4) | 49.2(+0.7) | 52.9(+1.1) |
| $P O_K$ | 42.4(+27.8) | 47.3(+46.1) | 52.0(+51.4) |
| $\mathrm{Ortho}(P'_h)$ | 42.6(+27.6) | 45.7(+44.5) | 51.8(+51.2) |

**Measuring the coding rate and robustness.** On the ViT-L variant of DEPICT-SA with an input resolution of $512 \times 512$, we measure the projected coding rate of image embeddings onto the subspaces spanned by $P'_h$ across layers, as shown in Figure 6. We find that the sign of the learned step size distinguishes two types of subspaces (distinguished by red and blue regions in the figure): one onto which the projected coding rate increases and the other onto which it decreases, which is consistent with our derivations. Furthermore, we measure the robustness of DEPICT under four types of parameter perturbations and find that DEPICT is surprisingly robust whereas Segmenter collapses, as shown in Table 4. We attribute the robustness of DEPICT to its awareness of and modeling for low-dimensional subspaces, which are not significantly altered under the parameter perturbations. Actually, the seminal work [36] has already noticed that it is the subspace, rather than individual units, that contains the semantic information in the high layers of neural networks. Our work demonstrates that this intriguing property can be strengthened by improving model designs, which yields better robustness.

## 5    Conclusion and Future Work

We proposed a compression perspective to view the Transformer decoders widely adopted in Transformer-based semantic segmentation, where we expect that class embeddings are actually the principal directions and thus they span the principal subspace and segment images via PCA. In experiments, we found that the principal subspace of the generic features extracted by the encoder is not ideal and that the principal directions are not flexible enough to align well with predefined classes. To this end, we extended the objectives of PCA to construct an ideal principal subspace and to find a low-rank approximation of image embeddings as classifiers. By unrolling the optimization procedure of these objectives, we derived a family of fully attentional white-box decoders, called DEPICT, providing theoretical interpretations for the empirical designs of the Transformer decoders. Experiments conducted on ADE20K have shown that DEPICT consistently outperforms its black-box counterpart, Segmenter, using significantly fewer parameters and FLOPs. We further validated the effectiveness of DEPICT on Cityscapes and Pascal Context datasets and investigated that DEPICT possesses desirable properties, such as orthogonality and robustness, as we expected and derived.

We believe that our work serves as a promising first step toward developing a comprehensive interpretation framework for Transformer-based semantic segmentation, and further efforts are needed to contribute to this goal. Focusing on interpretability, we use a relatively simple implementation for DEPICT from architecture designing to training tricks, compared to state-of-the-art methods. Whether the improvements for black-box models, such as hierarchical transformer encoders, pixel decoders, the mask classification formulation, and masked attention, are compatible with DEPICT remains an open question to explore.

## Acknowledgments and Disclosure of Funding

The authors would like to thank the constructive comments from anonymous reviewers. This work was partially supported by the National Natural Science Foundation of China under Grant 61876022. C.-G. Li is the corresponding author.

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

# Appendix

## A  Further Discussion on DEPICT

In this section, we aim to reiterate points that may lead to confusion, including subspaces and attention operators. Then we move on to discuss the prospects and challenges for generalizing DEPICT to the mask classification formulation.

### A.1  Reiterating All Mentioned Subspaces

Before diving into the MSSA operator, we would like to clarify several subspaces mentioned in our work. We referenced "a union of orthogonal subspaces" and its connections to LDR and MCR$^2$ in Section 2 and Section 3.2. Then we expected in Section 3.5 that, although our derivations target a single image, the extracted class embeddings will lie within a union of orthogonal subspaces on the scale of the entire dataset, and observed it indeed occurs in Section 4.2. Each of these subspaces corresponds to or represents a predefined class in our work and [43, 4].

We argue that the principal subspace of $Z_0$ is not ideal and assume an ideal principal subspace spanned by $P$ in Section 3.3. Thus, there are two types of principal subspace as image embeddings are refined: the current principal subspace of $Z_\ell$ and the ideal principal subspace we desire, until they converge on $Z_{L_1}$. Moreover, in Section 3.5, we mentioned "multiple principal subspaces" on the scale of the entire dataset, indicating that each image corresponds to a distinct principal subspace.

In Section 3.3, we propose to maximize the projected coding rate onto $P_h$, which consists of $M$ bases of the ideal principal subspace. As we generalize the derivation to the entire dataset scale in Section 3.5, $P_h$ actually consists of $M$ of bases of the "multiple principal subspaces".

### A.2  Differences among Self-Attention Operators

We discuss the differences among three types of self-attention operators: MHSA, the original MSSA, and our modified MSSA. There are four parameter matrices in MHSA. Nevertheless, due to parameter sharing, our MSSA has only one parameter matrix, $P$, and a learnable scalar, $\alpha$, as the step size. The original MSSA, however, adopts an additional learnable parameter matrix for simplicity of implementation [42]. For image classification, MSSA-based models lag slightly behind MHSA [42, 40], whereas our work demonstrates its superiority for Transformer-based semantic segmentation. Following the idea of MCR$^2$, despite decoupling its heads from classes, the original MSSA still models a homogeneous subspace in each head; in other words, these heads still correspond to some more basic or finer-grained concepts than classes. However, each head of our MSSA models a heterogeneous subspace, since its bases are likely related to several different classes.

### A.3  Generalizing to Mask Classification

For simplicity of analysis and implementation, our work focus on class embeddings, rather than the more general case of mask or instance embeddings proposed by [8]. This couples each class with merely one principal direction within an image. However, an image typically contains a small part of all predefined classes and each class allows rich intra-class variance, thus requiring more than one principal directions to represent itself. Therefore, it is rather natural to generalize to mask classification on the basis of our interpretations. Moreover, it is desirable to segment images at a finer granularity than patches—down to the pixel level—by similarly adopting a pixel decoder. However, this requires interactions among features at different scales, whereas our current work is limited to non-hierarchical features.

## B  Implementation Details

**Implementations of derived operators.** As we perform Layer Normalization (LN), which is used to normalize representations and then learn to scale them, before all attention operations in DEPICT, the scaling operations within our derived operators can be simplified. Taking MSSA as an example,

similarly for MSCA, it is implemented as follows:

$$\boldsymbol{Z}^{\ell+1} = \boldsymbol{Z}^\ell - \alpha \cdot \text{MSSA}(\boldsymbol{Z}^\ell \mid \boldsymbol{P}), \tag{22}$$

$$\text{MSSA}(\boldsymbol{Z} \mid \boldsymbol{P}) \doteq [\boldsymbol{P}'_1, \ldots, \boldsymbol{P}'_H] \begin{bmatrix} \text{SSA}(\boldsymbol{Z} \mid \boldsymbol{P}'_1) \\ \vdots \\ \text{SSA}(\boldsymbol{Z} \mid \boldsymbol{P}'_H) \end{bmatrix}, \tag{23}$$

$$\text{SSA}(\boldsymbol{Z} \mid \boldsymbol{P}'_h) \doteq (\boldsymbol{P}'_h{}^\top \boldsymbol{Z})\text{softmax}\left((\boldsymbol{P}'_h{}^\top \boldsymbol{Z})^\top (\boldsymbol{P}'_h{}^\top \boldsymbol{Z})\right), \quad \text{for} \quad h \in \{1, \ldots, H\}, \tag{24}$$

where $\alpha$ is actually a scaled step size.

**Variant settings.** For DEPICT-SA on the ADE20K dataset, we used three MSSA layers and set #heads×dim_head to 3×100 across all variants, with the exception of six MSSA layers for the ViT-L variant. For DEPICT-CA, we used three MSSA layers followed by three MSCA layers, and set #heads×dim_head to 3×100 in MSSA and #heads×dim_head to 3×50 in MSCA, across all variants, with the exception of setting 3×50 in MSSA for the ViT-S variant and six MSSA layers for the ViT-L variant. Only ViT-L variants of DEPICT-SA are evaluated on Cityscapes and Pascal Context, using two and four MSSA layers, and setting #heads×dim_head to 4×19 and 4×60, respectively.

## C  Additional Experiments

**Investigating the relative relationship between $R(\boldsymbol{Z})$ and $R(\boldsymbol{Q})$.** To make the visualization more intuitive, we measure $R(\boldsymbol{Q})$, which is larger than $R(\overline{\boldsymbol{Q}})$ since that an image contains only a small part of all the classes. Thus, $R(\boldsymbol{Q})$ serves as an upper bound of $R(\overline{\boldsymbol{Q}})$. As shown in Figure 7, $R(\boldsymbol{Q})$ is indeed smaller than $R(\boldsymbol{Z})$ and the difference between them roughly decreases as the depth increases.

**Ablation studies.** As shown in Table 5, a relatively small number of heads produces the best performance, implying that an over-tight lower bound may make training less effective. In [48] it is also observed that the performance initially increases with the number of heads, then declines.

**Visualization of more variants.** We present the visualization of the inner product of the parameter vectors and class embeddings for all variants trained on ADE20K in Figures 9 and 10. We also provide additional segmentation results via PCA and compare them to those from DEPICT in Figure 8.

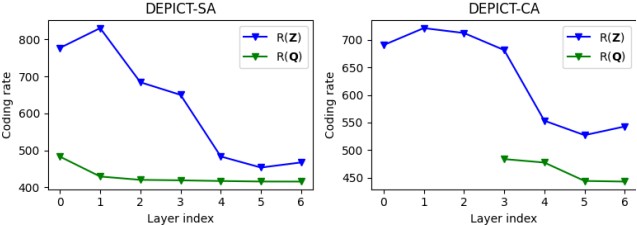

**Figure 7: Measuring $R(\boldsymbol{Z})$ and $R(\boldsymbol{Q})$ across layers.** All are based on ViT-L with an input resolution of 640×640.

## D  Relevant Proofs

### D.1  Lower and Upper Bounds for the Projected Coding Rate

Given image embeddings $\boldsymbol{Z} \in \mathbb{R}^{D \times N}$, and an $M$-dimensional subspace spanned by a set of orthonormal bases $\boldsymbol{P}' \in \mathbb{R}^{D \times M}$, the projected coding rate of $\boldsymbol{Z}$ onto $\boldsymbol{P}'$ is:

$$R(\boldsymbol{P}'^\top \boldsymbol{Z}; \epsilon) = \frac{1}{2} \log \det(\boldsymbol{I}_M + \frac{M}{N\epsilon^2} (\boldsymbol{P}'^\top \boldsymbol{Z})(\boldsymbol{P}'^\top \boldsymbol{Z})^\top). \tag{25}$$

Specifically, the projected coding rate onto one of the bases of the subspace, say $\boldsymbol{p}'_i$, is

$$R(\boldsymbol{p}_i'^\top \boldsymbol{Z}; \epsilon) = \frac{1}{2} \log \det(1 + \frac{1}{N\epsilon^2} (\boldsymbol{p}_i'^\top \boldsymbol{Z})(\boldsymbol{p}_i'^\top \boldsymbol{Z})^\top). \tag{26}$$

**Table 5: Ablation studies.** *Left*: All are based on ViT-S and compared to the naive implementation which adopts MSSA and ISTA, with setting #heads×dim_head to 3×50, #layer to 2 and dropout to 0.0. *Right*: Investigating the impact of #heads and dim_head.

| variants | mIoU | #params |
|---|---|---|
| MSSA + ISTA | 45.6 | 0.47M |
| #heads = 1 | 45.4(-0.2) | 0.47M |
| #heads = 5 | 45.6(+0.0) | 0.47M |
| #layers = 1 | 45.2(-0.4) | 0.27M |
| #layers = 3 | 45.8(+0.2) | 0.68M |
| dropout = 0.1 | 45.2(-0.4) | 0.47M |
| MSSA only | 45.4(-0.2) | 0.18M |
| ISTA only | 44.9(-0.7) | 0.36M |
| MSSA + MLP | 46.2(+0.6) | 2.5M |
| MHSA + ISTA | 44.9(-0.7) | 1.5M |
| MHSA + MLP (Segmenter) | 45.3(-0.3) | 4.1M |
| **DEPICT-SA** | 46.7(+1.1) | 0.41M |

| #head×dim_head | mIoU |
|---|---|
| ViT-S Backbone | |
| 1×300 | 45.0 |
| 2×150 | 45.7 |
| 3×100 | 46.7 |
| 4×75 | 46.3 |
| 1×384 | 46.2 |
| ViT-B Backbone | |
| 3×50 | 48.8 |
| 3×100 | 49.3 |
| 3×150 | 49.4 |
| 3×200 | 49.4 |
| 3×250 | 49.5 |

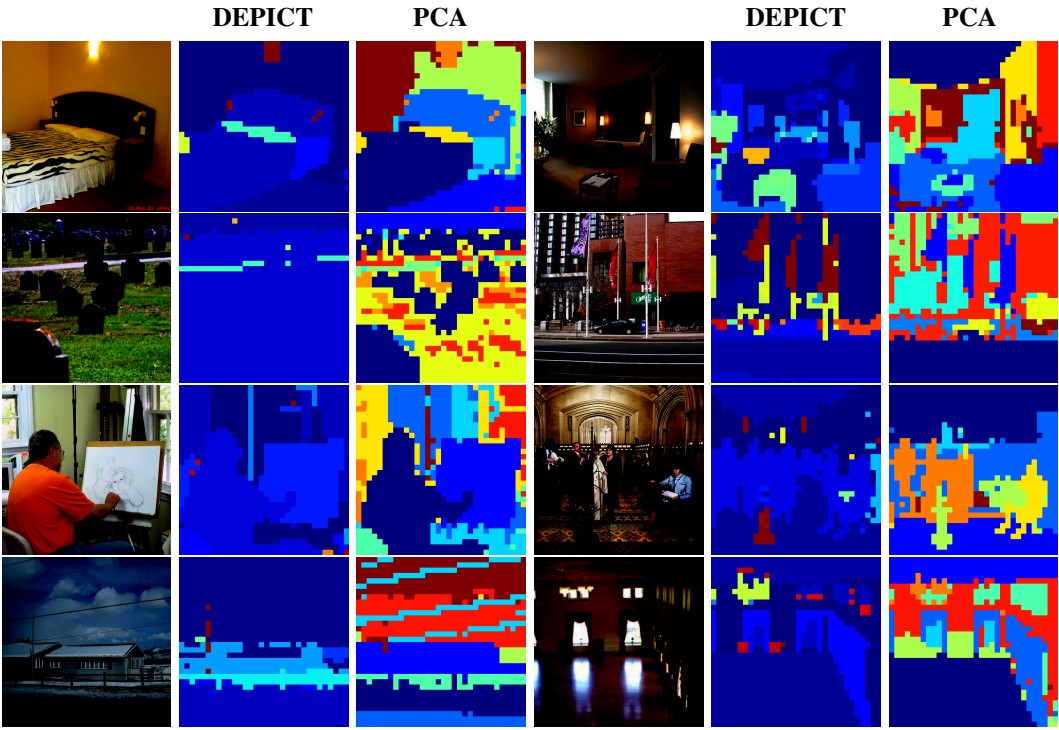

**Figure 8: Image segmentation via PCA and DEPICT.** Given an image, we segment it via DEPICT and PCA, respectively. We perform PCA on the representations $Z_{L_1}$ and use the first 10 principal directions as cluster centroids. We find that, with an ideal principal subspace constructed, PCA serves as a surprisingly effective method for image segmentation, and it captures additional information compared to DEPICT, which is trained in an end-to-end manner. For example, numbering the images in subfigure row-wise from a to f, it differentiates two different walls (see subfigure a); tombstones and grass (see subfigure c); sitting and standing persons (see subfigure f); sky, cloud, and power lines (see subfigure g). However, as an unsupervised method, PCA still does not align well with supervision, especially in complicated scenarios (see subfigure b and d).

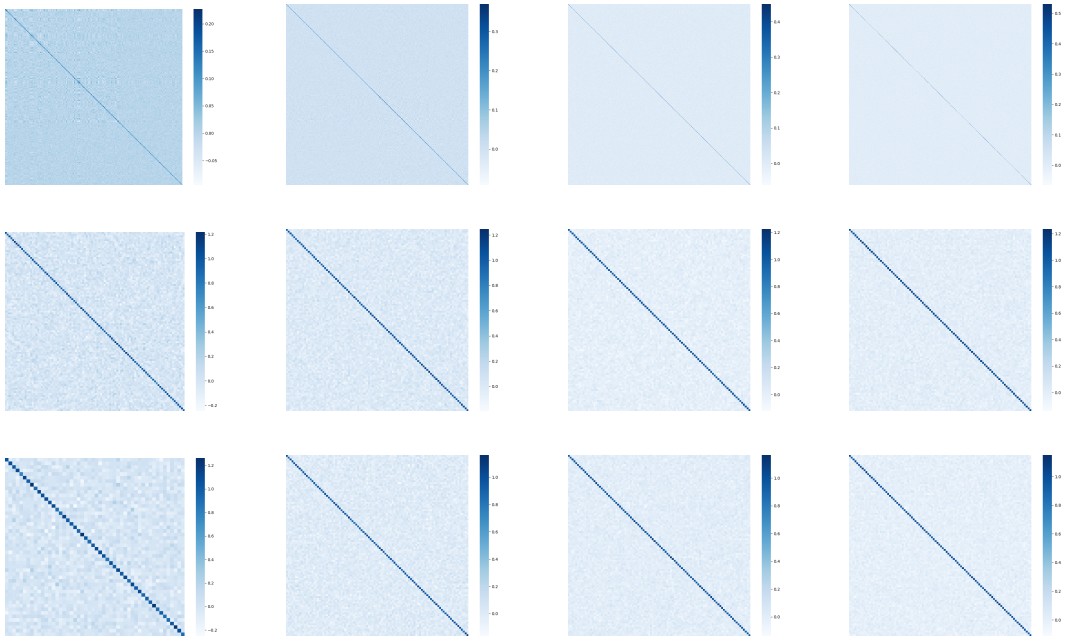

**Figure 9: Visualization of the inner product of parameter vectors.** From top to bottom, each row is based on Segmenter, DEPICT-SA, and DEPICT-CA, respectively; from left to right, each column is based on ViT-T, ViT-S, ViT-B, and ViT-L, respectively.

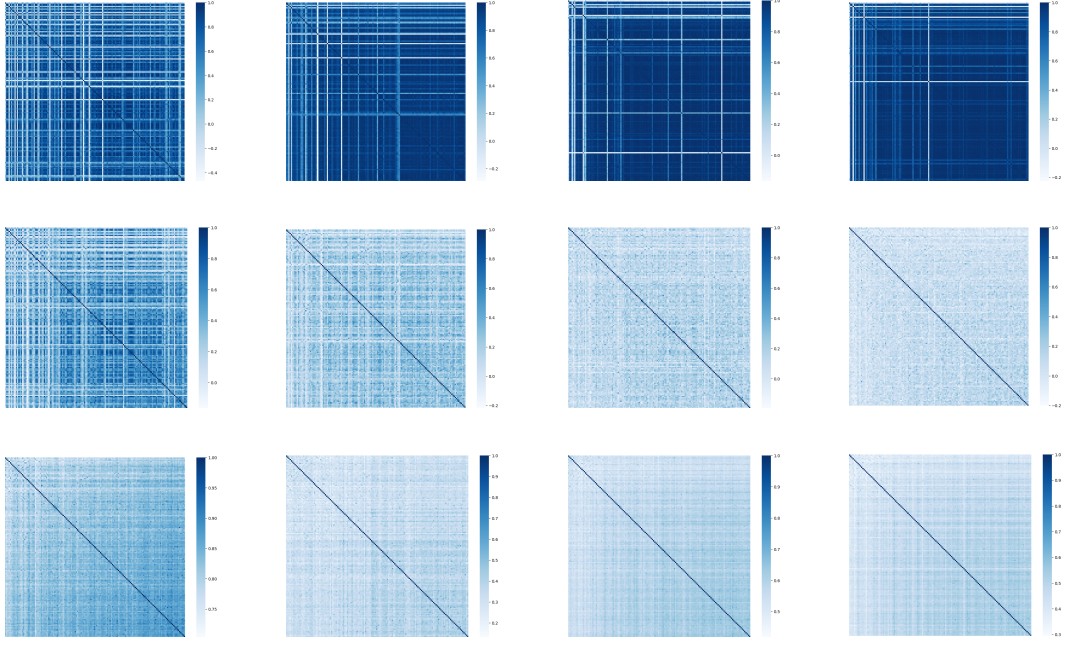

**Figure 10: Visualization of the inner product of class embeddings.** From top to bottom, each row is based on Segmenter, DEPICT-SA, and DEPICT-CA, respectively; from left to right, each column is based on ViT-T, ViT-S, ViT-B, and ViT-L, respectively.

By Lemma A.2 of [43], the projected coding rate in Eq. (25) can also be calculated as

$$R(\boldsymbol{P}'^\top \boldsymbol{Z}; \epsilon) = \frac{1}{2} \log \det(\boldsymbol{I}_M + \frac{M}{N\epsilon^2} (\boldsymbol{P}'^\top \boldsymbol{Z})^\top (\boldsymbol{P}'^\top \boldsymbol{Z})), \tag{27}$$

which is equivalent to

$$\frac{1}{2} \log \det(\boldsymbol{I}_M + \frac{M}{N\epsilon^2} \boldsymbol{Z}^\top \boldsymbol{P}' \boldsymbol{P}'^\top \boldsymbol{Z}) \tag{28}$$

$$= \frac{1}{2} \log \det(\boldsymbol{I}_M + \frac{M}{N\epsilon^2} \boldsymbol{Z}^\top [\boldsymbol{p}'_1, \ldots, \boldsymbol{p}'_M] \begin{bmatrix} \boldsymbol{p}'^\top_1 \\ \vdots \\ \boldsymbol{p}'^\top_M \end{bmatrix} \boldsymbol{Z}) \tag{29}$$

$$= \frac{1}{2} \log \det(\boldsymbol{I}_M + \frac{M}{N\epsilon^2} \boldsymbol{Z}^\top \left( \sum_{i=1}^{M} \boldsymbol{p}_i' \boldsymbol{p}_i'^\top \right) \boldsymbol{Z}) \tag{30}$$

$$= \frac{1}{2} \log \det \left( \frac{1}{M} \left( \sum_{i=1}^{M} \boldsymbol{I}_M + \frac{M^2}{N\epsilon^2} \boldsymbol{Z}^\top \boldsymbol{p}_i' \boldsymbol{p}_i'^\top \boldsymbol{Z} \right) \right). \tag{31}$$

By Lemma A.1 of [43], the right-hand side of (31) is bounded below by

$$\frac{1}{2M} \sum_{i=1}^{M} \log \det(\boldsymbol{I}_M + \frac{M^2}{N\epsilon^2} \boldsymbol{Z}^\top \boldsymbol{p}_i' \boldsymbol{p}_i'^\top \boldsymbol{Z}). \tag{32}$$

In other words, we have shown that

$$R(\boldsymbol{P}'^\top \boldsymbol{Z}; \epsilon) \geq \frac{1}{M} \sum_{i=1}^{M} R(\frac{\boldsymbol{p}_i'^\top \boldsymbol{Z}}{M}; \epsilon) = \frac{1}{M} \sum_{i=1}^{M} R(\boldsymbol{p}_i'^\top \boldsymbol{Z}; M\epsilon). \tag{33}$$

By the proof of Lemma A.4 in [43], we have that

$$\log \det(\boldsymbol{X}) \leq \log \det(\boldsymbol{Y}) + \mathrm{trace}(\boldsymbol{Y}^{-1}\boldsymbol{X}) - MN, \quad \text{for all} \quad \{\boldsymbol{X}, \boldsymbol{Y}\} \subseteq \mathbb{S}_{++}^{MN}, \tag{34}$$

where $\mathbb{S}_{++}^{MN}$ denotes the set of positive symmetric matrices of size $MN \times MN$. We take the specific values for $\boldsymbol{X}$ and $\boldsymbol{Y}$ as follows:

$$\boldsymbol{Y} = \boldsymbol{I}_{MN} + \frac{1}{N\epsilon^2} \begin{bmatrix} \boldsymbol{Z}^\top \boldsymbol{p}'_1 \boldsymbol{p}'^\top_1 \boldsymbol{Z} & \boldsymbol{0} & \ldots & \boldsymbol{0} \\ \boldsymbol{0} & \boldsymbol{Z}^\top \boldsymbol{p}_2' \boldsymbol{p}_2'^\top \boldsymbol{Z} & \ldots & \boldsymbol{0} \\ \vdots & \vdots & \ddots & \vdots \\ \boldsymbol{0} & \boldsymbol{0} & \ldots & \boldsymbol{Z}^\top \boldsymbol{p}'_M \boldsymbol{p}'^\top_M \boldsymbol{Z} \end{bmatrix}, \tag{35}$$

$$\boldsymbol{X} = \boldsymbol{I}_{MN} + \frac{M}{N\epsilon^2} \begin{bmatrix} \boldsymbol{Z}^\top (\sum_{i=1}^{M} \boldsymbol{p}_i' \boldsymbol{p}_i'^\top) \boldsymbol{Z} & \boldsymbol{0} & \ldots & \boldsymbol{0} \\ \boldsymbol{0} & \boldsymbol{0} & \ldots & \boldsymbol{0} \\ \vdots & \vdots & \ddots & \vdots \\ \boldsymbol{0} & \boldsymbol{0} & \ldots & \boldsymbol{0} \end{bmatrix}. \tag{36}$$

By the property of determinant for block diagonal matrix, we have

$$\log \det(\boldsymbol{Y}) = \sum_{i=1}^{M} \log \det(\boldsymbol{I}_N + \frac{1}{N\epsilon^2} (\boldsymbol{p}_i'^\top \boldsymbol{Z})(\boldsymbol{p}_i'^\top \boldsymbol{Z})^\top), \tag{37}$$

$$\log \det(\boldsymbol{X}) = \log \det(\boldsymbol{I}_N + \frac{M}{N\epsilon^2} \boldsymbol{Z}^\top (\sum_{i=1}^{M} \boldsymbol{p}_i' \boldsymbol{p}_i'^\top) \boldsymbol{Z}). \tag{38}$$

Moreover, we have that

$$\mathrm{trace}(\boldsymbol{Y}^{-1}\boldsymbol{X}) = \mathrm{trace}\left( \left( \boldsymbol{I}_N + \frac{1}{N\epsilon^2} \boldsymbol{Z}^\top \boldsymbol{p}'_1 \boldsymbol{p}'^\top_1 \boldsymbol{Z} \right)^{-1} \left( \boldsymbol{I}_N + \frac{M}{N\epsilon^2} \boldsymbol{Z}^\top \left( \sum_{i=1}^{M} \boldsymbol{p}_i' \boldsymbol{p}_i'^\top \right) \boldsymbol{Z} \right) \right)$$

$$+ \sum_{i=2}^{M} \mathrm{trace}\left( \left( \boldsymbol{I}_N + \frac{1}{N\epsilon^2} \boldsymbol{Z}^\top \boldsymbol{p}_i' \boldsymbol{p}_i'^\top \boldsymbol{Z} \right)^{-1} \right). \tag{39}$$

Without loss of generality, we assume that

$$
\begin{aligned}
\beta =\ &\mathrm{trace}\left(\left(\boldsymbol{I}_N + \frac{1}{N\epsilon^2}\boldsymbol{Z}^\top \boldsymbol{p}'_1 \boldsymbol{p}'^\top_1 \boldsymbol{Z}\right)^{-1}\right) \\
\geq\ &\mathrm{trace}\left(\left(\boldsymbol{I}_N + \frac{1}{N\epsilon^2}\boldsymbol{Z}^\top \boldsymbol{p}_i' \boldsymbol{p}_i'^\top \boldsymbol{Z}\right)^{-1}\right),\ \ \text{for}\ i \in \{2,\dots,M\},
\end{aligned}
\tag{40}
$$

then the quantity in (39) is upper bounded by

$$
\mathrm{trace}\left((\boldsymbol{I}_N + \frac{1}{N\epsilon^2}\boldsymbol{Z}^\top \boldsymbol{p}'_1 \boldsymbol{p}'^\top_1 \boldsymbol{Z})^{-1}(\boldsymbol{I}_N + \frac{M}{N\epsilon^2}\boldsymbol{Z}^\top (\sum_{i=1}^M \boldsymbol{p}_i' \boldsymbol{p}_i'^\top)\boldsymbol{Z})\right) + (M-1)\beta
\tag{41}
$$

$$
=\mathrm{trace}\left((\boldsymbol{I}_N + \frac{1}{N\epsilon^2}\boldsymbol{Z}^\top \boldsymbol{p}'_1 \boldsymbol{p}'^\top_1 \boldsymbol{Z})^{-1}(M\sum_{i=1}^M \boldsymbol{I}_N + \frac{1}{N\epsilon^2}\boldsymbol{Z}^\top \boldsymbol{p}_i' \boldsymbol{p}_i'^\top \boldsymbol{Z} - (M^2-1)\boldsymbol{I}_N)\right)
$$
$$
+(M-1)\beta
\tag{42}
$$

$$
=MN + M\sum_{i=2}^M \mathrm{trace}\left((\boldsymbol{I}_N + \frac{1}{N\epsilon^2}\boldsymbol{Z}^\top \boldsymbol{p}'_1 \boldsymbol{p}'^\top_1 \boldsymbol{Z})^{-1}(\boldsymbol{I}_N + \frac{1}{N\epsilon^2}\boldsymbol{Z}^\top \boldsymbol{p}_i' \boldsymbol{p}_i'^\top \boldsymbol{Z})\right) - M(M-1)\beta.
\tag{43}
$$

Without loss of generality, we assume that

$$
\begin{aligned}
&\mathrm{trace}\left((\boldsymbol{I}_N + \frac{1}{N\epsilon^2}\boldsymbol{Z}^\top \boldsymbol{p}'_1 \boldsymbol{p}'^\top_1 \boldsymbol{Z})^{-1}(\boldsymbol{I}_N + \frac{1}{N\epsilon^2}\boldsymbol{Z}^\top \boldsymbol{p}'_M \boldsymbol{p}'^\top_M \boldsymbol{Z})\right) \\
&\geq \mathrm{trace}\left((\boldsymbol{I}_N + \frac{1}{N\epsilon^2}\boldsymbol{Z}^\top \boldsymbol{p}'_1 \boldsymbol{p}'^\top_1 \boldsymbol{Z})^{-1}(\boldsymbol{I}_N + \frac{1}{N\epsilon^2}\boldsymbol{Z}^\top \boldsymbol{p}_i' \boldsymbol{p}_i'^\top \boldsymbol{Z})\right),\ i \in \{1,\dots,M\},
\end{aligned}
\tag{44}
$$

then (43) is upper bounded by

$$
MN + M(M-1)\mathrm{trace}\left((\boldsymbol{I}_N + \frac{1}{N\epsilon^2}\boldsymbol{Z}^\top \boldsymbol{p}'_1 \boldsymbol{p}'^\top_1 \boldsymbol{Z})^{-1}(\frac{1}{N\epsilon^2}\boldsymbol{Z}^\top \boldsymbol{p}'_M \boldsymbol{p}'^\top_M \boldsymbol{Z})\right)
\tag{45}
$$

$$
=MN + M(M-1)\mathrm{trace}\left((\boldsymbol{I}_N + \frac{1}{N\epsilon^2}(\boldsymbol{p}'^\top_1 \boldsymbol{Z})^\top(\boldsymbol{p}'^\top_1 \boldsymbol{Z}))^{-1}(\frac{1}{N\epsilon^2}(\boldsymbol{p}'^\top_M \boldsymbol{Z})^\top(\boldsymbol{p}'^\top_M \boldsymbol{Z}))\right).
\tag{46}
$$

Since that the rank-1 matrix $(\boldsymbol{p}'^\top_i \boldsymbol{Z})^\top(\boldsymbol{p}'^\top_i \boldsymbol{Z}) \in \mathbb{R}^{N\times N}$ has a single non-zero eigenvalue, i.e., $N\mathrm{Var}(\boldsymbol{p}'^\top_i \boldsymbol{Z}) = (\boldsymbol{p}'^\top_i \boldsymbol{Z})(\boldsymbol{p}'^\top_i \boldsymbol{Z})^\top$. By Ruhe's trace inequality [34], we have that (46) is upper bounded by

$$
MN + M(M-1)\frac{\mathrm{Var}(\boldsymbol{p}'^\top_M \boldsymbol{Z})}{\mathrm{Var}(\boldsymbol{p}'^\top_1 \boldsymbol{Z}) + \epsilon^2}
\tag{47}
$$

$$
\leq MN + M(M-1)\frac{\mathrm{Var}(\boldsymbol{p}'^\top_M \boldsymbol{Z})}{\mathrm{Var}(\boldsymbol{p}'^\top_1 \boldsymbol{Z})}.
\tag{48}
$$

We argue that since $\boldsymbol{P}'$ lies in the principal subspace in our case, the projected variance on different bases does not differ significantly; in other words, the ratio listed above is upper bounded by a constant with respect to $\boldsymbol{Z}$. We therefore use $\gamma$ to denote the product of $\frac{1}{2}M(M-1)$ and this constant. Now, by using (37), (38), (48) and (34), we derive

$$
R(\boldsymbol{P}'^\top \boldsymbol{Z};\epsilon) \leq \sum_{i=1}^M R(\boldsymbol{p}'^\top_i \boldsymbol{Z};\epsilon) + \gamma.
\tag{49}
$$

In summary, we have proved that

$$
\frac{1}{M}\sum_{i=1}^M R(\boldsymbol{p}'^\top_i \boldsymbol{Z};M\epsilon) \leq R(\boldsymbol{P}'^\top \boldsymbol{Z};\epsilon) \leq \sum_{i=1}^M R(\boldsymbol{p}'^\top_i \boldsymbol{Z};\epsilon) + \gamma.
\tag{50}
$$

## D.2 Evaluate the Quality of Solutions for Low-Rank Approximation

In this section, we evaluate the quality of the solutions, specifically the principal directions and the cluster centroids $k$-means, for the low-rank approximation problem measured by the coding rate defined in Eq. (14). The coding rate of $Z$ and $\overline{Q}$ is given by

$$R(Z) = \frac{1}{2}\log\det(I_D + \frac{D}{N\epsilon^2}ZZ^\top), \tag{51}$$

$$R(\overline{Q}) = \frac{1}{2}\log\det(I_D + \frac{D}{N\epsilon^2}\overline{Q}\overline{Q}^\top). \tag{52}$$

Specifically, we assume that $Q$ consists of the leading $C$ principal directions.

**Evaluating principal directions.** Letting there be $n_i$ instances of $q_i$ in $\overline{Q}$, where $\sum_1^C n_i = N$, we have

$$\overline{Q}\overline{Q}^\top = \sum_{i=1}^C n_i q_i q_i^\top \tag{53}$$

$$= Q\mathrm{diag}\{n_1,\ldots,n_C\}Q^\top \tag{54}$$

$$= (Q\mathrm{diag}\{n_1^{\frac{1}{2}},\ldots,n_C^{\frac{1}{2}}\})(Q\mathrm{diag}\{n_1^{\frac{1}{2}},\ldots,n_C^{\frac{1}{2}}\})^\top. \tag{55}$$

By Lemma A.2 of [43], when calculating the coding rate of $\overline{Q}$, we can rewrite (55) as follows:

$$(Q\mathrm{diag}\{n_1^{\frac{1}{2}},\ldots,n_C^{\frac{1}{2}}\})^\top(Q\mathrm{diag}\{n_1^{\frac{1}{2}},\ldots,n_C^{\frac{1}{2}}\}) \tag{56}$$

$$=\mathrm{diag}\{n_1^{\frac{1}{2}},\ldots,n_C^{\frac{1}{2}}\}Q^\top Q\mathrm{diag}\{n_1^{\frac{1}{2}},\ldots,n_C^{\frac{1}{2}}\}. \tag{57}$$

Since that the principal directions are orthonormal, Eq. (57) turns out to be $\mathrm{diag}\{n_1,\ldots,n_C\}$. By using PCA, the principal directions are essentially the eigenvector of $ZZ^\top$, i.e.,

$$ZZ^\top Q = Q\mathrm{diag}\{\lambda_1,\ldots,\lambda_C\}, \tag{58}$$

where $\lambda_1 > \ldots > \lambda_C$ is the leading $C$ eigenvalues of $ZZ^\top$. Therefore, as long as each $n_i$ is sufficiently close to $\lambda_i$, $i \in \{1,\ldots,C\}$, the principal directions are a sufficiently good solution. By taking a transpose on both sides of (58) and multiplying by $Q$, we get

$$Q^\top ZZ^\top Q = \mathrm{diag}\{\lambda_1,\ldots,\lambda_C\}Q^\top Q = \mathrm{diag}\{\lambda_1,\ldots,\lambda_C\}. \tag{59}$$

We find that $\lambda_i$ is actually the variance projected onto $q_i$ multiplied by $N$. Thus, we conclude that the more discriminative the principal directions are, the better the solutions. In extreme cases, when all embeddings in each class are identical to a certain principal direction, the principal directions are the optimal solution.

**Evaluating $k$-means cluster centroids.** Given the cluster membership indicators for $k$-means clustering defined in [13], i.e., $H_C = [h_1,\ldots,h_C] \in \mathbb{R}^{N\times C}$, the cluster centroids are

$$V = Z[\frac{h_1}{\sqrt{n_1}},\ldots,\frac{h_C}{\sqrt{n_C}}] \tag{60}$$

$$= ZH_C\mathrm{diag}^{-1}\{n_1^{\frac{1}{2}},\ldots,n_C^{\frac{1}{2}}\} \tag{61}$$

where $h_c$ consists of $n_c$ 1's, with all other entries being zero. Then we have

$$V\mathrm{diag}\{n_1^{\frac{1}{2}},\ldots,n_C^{\frac{1}{2}}\}T = ZH_C T = ZW, \tag{62}$$

$$W \doteq H_C T, \tag{63}$$

where $T \in \mathbb{R}^{C\times C}$ is an orthogonal matrix whose last column is $(\sqrt{n_1/N},\ldots,\sqrt{n_C/N})^\top$. By Theorem 3.1 of [13], we have

$$V\mathrm{diag}\{n_1^{\frac{1}{2}},\ldots,n_C^{\frac{1}{2}}\}T = ZW = [ZZ^\top q_1/\lambda_1^{\frac{1}{2}},\ldots,ZZ^\top q_{C-1}/\lambda_{C-1}^{\frac{1}{2}},\mathbf{0}] \tag{64}$$

$$= [\lambda_1^{\frac{1}{2}}q_1,\ldots,\lambda_{C-1}^{\frac{1}{2}}q_{C-1},\mathbf{0}]. \tag{65}$$

Therefore, we have

$$\boldsymbol{V} = [\lambda_1^{\frac{1}{2}}\boldsymbol{q}_1, \ldots, \lambda_{C-1}^{\frac{1}{2}}\boldsymbol{q}_{C-1}, \boldsymbol{0}]\boldsymbol{T}^\top \mathrm{diag}^{-1}\{n_1^{\frac{1}{2}}, \ldots, n_C^{\frac{1}{2}}\}, \tag{66}$$

which bridges the principal directions and the centroids of the $k$ mean clusters. Then for the same reason of (55) and (57), we have

$$\overline{\boldsymbol{V}}^\top \overline{\boldsymbol{V}} = \mathrm{diag}\{\lambda_1, \ldots, \lambda_{C-1}, 0\}. \tag{67}$$

We conclude that the better the image embeddings fit a $(C-1)$-dimensional subspace, the more optimal the $k$-means cluster centroids become. In the extreme case where the image embeddings lie perfectly within a $(C-1)$-dimensional subspace, the $k$-means cluster centroids are the optimal solution.

### D.3 Understanding self-attention applied to the concatenation of $Z$ and $Q$

Given image embeddings $\boldsymbol{Z} \in \mathbb{R}^{D \times N}$ and class embeddings $\boldsymbol{Q} \in \mathbb{R}^{D \times C}$, we concatenate them as $\overline{\boldsymbol{Z}} = [\boldsymbol{Z}, \boldsymbol{Q}]$ and then, without loss of generality, conduct non-parametric self-attention on it, i.e.,

$$\overline{\boldsymbol{Z}} \cdot \mathrm{softmax}\left(\overline{\boldsymbol{Z}}^\top \overline{\boldsymbol{Z}}\right) = [\boldsymbol{Z}, \boldsymbol{Q}] \cdot \mathrm{softmax}\left(\begin{bmatrix} \boldsymbol{Z}^\top \boldsymbol{Z} & \boldsymbol{Z}^\top \boldsymbol{Q} \\ \boldsymbol{Q}^\top \boldsymbol{Z} & \boldsymbol{Q}^\top \boldsymbol{Q} \end{bmatrix}\right). \tag{68}$$

To simplify the analysis, we remove the *softmax* operator. As softmax computes along the last dimension (i.e., each column) by default, it essentially scales the update of each embedding. And since we interpret attention as a gradient step, this scaling does not alter the corresponding objective. We thus have

$$\overline{\boldsymbol{Z}}^{\ell+1} = \overline{\boldsymbol{Z}}^\ell - \alpha[\boldsymbol{Z}^\ell, \boldsymbol{Q}^\ell]\begin{bmatrix} \boldsymbol{Z}^{\ell\top}\boldsymbol{Z}^\ell & \boldsymbol{Z}^{\ell\top}\boldsymbol{Q}^\ell \\ \boldsymbol{Q}^{\ell\top}\boldsymbol{Z}^\ell & \boldsymbol{Q}^{\ell\top}\boldsymbol{Q}^\ell \end{bmatrix}, \tag{69}$$

$$[\boldsymbol{Z}^{\ell+1}, \boldsymbol{Q}^{\ell+1}] = [\boldsymbol{Z}^\ell, \boldsymbol{Q}^\ell] - \alpha[\boldsymbol{Z}^\ell\boldsymbol{Z}^{\ell\top}\boldsymbol{Z}^\ell + \boldsymbol{Q}^\ell\boldsymbol{Q}^{\ell\top}\boldsymbol{Z}^\ell, \boldsymbol{Z}^\ell\boldsymbol{Z}^{\ell\top}\boldsymbol{Q} + \boldsymbol{Q}^\ell\boldsymbol{Q}^{\ell\top}\boldsymbol{Q}^\ell] \tag{70}$$

$$= [\boldsymbol{Z}^\ell - \alpha(\boldsymbol{Z}^\ell\boldsymbol{Z}^{\ell\top}\boldsymbol{Z}^\ell + \boldsymbol{Q}^\ell\boldsymbol{Q}^{\ell\top}\boldsymbol{Z}^\ell), \boldsymbol{Q}^\ell - \alpha(\boldsymbol{Z}^\ell\boldsymbol{Z}^{\ell\top}\boldsymbol{Q} + \boldsymbol{Q}^\ell\boldsymbol{Q}^{\ell\top}\boldsymbol{Q}^\ell)]. \tag{71}$$

In Eq. (71), there are four gradient terms: the terms $\boldsymbol{Z}^\ell\boldsymbol{Z}^{\ell\top}\boldsymbol{Z}^\ell$ and $\boldsymbol{Q}^\ell\boldsymbol{Q}^{\ell\top}\boldsymbol{Q}^\ell$ represent self-attention, and the terms $\boldsymbol{Z}^\ell\boldsymbol{Z}^{\ell\top}\boldsymbol{Q}$ represent cross-attention. These have all been discussed in Section 3.3 and Section 3.4. However, the objective corresponding to $\boldsymbol{Q}^\ell\boldsymbol{Q}^{\ell\top}\boldsymbol{Z}^\ell$ remains unknown to us.

