# OpenReview forum: "Rethinking Decoders for Transformer-based Semantic Segmentation: A Compression Perspective"
_NeurIPS.cc/2024/Conference — NeurIPS 2024 poster_

### Official Review · Reviewer_V3V2 · 2024-07-10

**Soundness:** 3
**Presentation:** 2
**Contribution:** 3
**Rating:** 5
**Confidence:** 3

**Summary:**

This paper introduces a novel perspective by conceptualizing semantic segmentation as data compression, akin to PCA, which simplifies the role of decoders in Transformer-based models. It presents DEPICT, white-box decoders that clarify the functions of learnable class embeddings, self-attention and dot-product operations, achieving comparable or better performance with fewer parameters on the ADE20K dataset compared to traditional methods. This work enhances the understanding of decoder mechanisms in semantic segmentation.

**Strengths:**

1. The figures and tables in the paper are informative.
2. The experiments validate that the proposed methods reduce model parameters while achieving adequate performance.
3. The paper provides thorough mathematical derivations of the related theories.

**Weaknesses:**

1. The experimental validation is limited as it only involves a single dataset, ADE20K. Expanding the testing to include a variety of datasets along with a detailed analysis of FLOPs would provide a stronger evidence of the methods' efficiency and adaptability.

2. Some mathematical proofs are confusing or inadequately explained:
   - **Line 144**: The assertion of strict equivalence lacks a thorough explanation, although it may seem intuitively correct.
   - **Equation (4)**: The text does not clearly explain the implications of transposing elements $(p_c^T Z)^\top$ and $(p_c^T Z)$, especially in terms of how the unit matrix is affected, which could lead to dimensional inconsistencies.
   - **Line 412**: The equivalence between minimizing $|log(1 + \beta)|$ where $\beta > 0$ and minimizing $|\beta – 1|$ is confusing, as logically the former minimizes when $\beta$ approaches zero, whereas the latter minimizes when $\beta$ equals one.

3. Figure 2's subfigures (b) and (c) appear identical apart from font color, and the caption provides no explanation, raising questions about whether this is an oversight or intended to convey a specific message.

4. The claim on Line 50 that semantic segmentation can be viewed as PCA is too broad and not specific to the novel contributions of this paper, as it could apply broadly to any task involving classification.

**Questions:**

1. The main concern is the brevity of the experimental section, which focuses solely on the ADE20K dataset. It is necessary to expand experimental validation to include a variety of datasets and different model architectures to thoroughly evaluate the generalizability and robustness of the proposed methods.

2. The proofs within the paper are notably difficult to comprehend, and due to specific issues in detail and explanation as mentioned, their rigor cannot be assured. This ambiguity affects the verification of their correctness. It is crucial for the authors to not only address the aforementioned concerns but also to refine the presentation of these proofs to enable readers to effectively assess their accuracy.

**Limitations:**

The paper acknowledges its limitations in the appendix. There is no discussion of potential negative societal impacts, as they are not applicable to this technical research.

---

> ### Author Rebuttal · Authors · 2024-08-07
>
> We highly appreciate your valuable comments, especially for your attention to the details in our paper.
>
> **Response to Weakness #1.** We conduct more experiment and provide the new experimental results on Cityscapes and Pascal Context (in Common Response).  Due to time limitation, we did not provide ViT-Large based DEPICT on these two datasets (at this moment), but we will get it done soon (maybe during the discussion or at least in the final version). Based on ViT-Small, we show that our DEPICT outperforms Segmenter by 7% mIoU on both Cityscapes and Pascal Context. Regarding the GFLOPs metric, our decoder requires remarkably fewer GFLOPs as it is fully attention based method and utilizes the MSSA block. The GFLOPs of our decoder is 2/3/4/10 based on ViT-Tiny/Small/Base/Large, whereas the segmenter's is 2/6/22/37.
>
> **Response to Weakness #2.** The conclusion of line 144 is justified by that maximizing a scalar $x$ is equivalent to maximizing $\log(1+\alpha x)$ where $0 \leq x \leq 1$, $\alpha > 0$. The identity matrix of equation (3) degenerate to scalar 1, and the identity  matrix of equation (4) has a shape of $N \times N$, where $N$ is the sequence length. Despite the dimensions are inconsistent, equation (3) is strictly equivalent to equation (4), by Lemma A.2 of [1]. For line 412, it is indeed a mistake, which should be corrected as "being equivalent to minimizing $|\beta|$". Thanks for pointing it! Fortunately, this mistake doesn't affect the correctness of the following equation (37). We are willing to checking or refining our proofs. For example, we have proposed much more concise and intuitive proofs from the perspective of low-rank approximation (see Response to Question#2).
>
> **Response to Weakness #3.** It is our fault for providing no explanation about the font color. The red color means that the parameters are somewhat constrained as we constrained that $\text{heads}*\text{dim}$_$\text{head}=C$. However, as explained in our Common Response, we now remove this constraint, as it only holds for a single image rather than for the whole dataset.
>
> **Response to Weakness #4.** This is a very insightful concern. A model for semantic segmentation processes all the patch embeddings of a singe image and tries to classify each of them, i.e., both the transformation and the classification are done at the patch-level. Such consistency enables us to interpret it by the idea of PCA. In other words, as models are typically only allowed to make direct information exchange within pixels or patches of a singe image, image segmentation is the most suitable task for applying PCA. Although based on semantic segmentation, our proposed interpretation framework can be applied to other dense prediction tasks. For example, we can allow that a class can be represented by more than one principal directions and each of these principal directions stand for an object or a segment, resulting in object detection and instance segmentation. However, it is challenging to adopt our interpretation from pixel-level tasks to image-level tasks. But, we believe that compression is an inevitable perspective for many tasks and our work serves as a first yet a solid step toward developing a comprehensive interpretation framework.
>
> **Response to Question #1.** Please refer to Response to Weakness #1.
>
> **Response to Question #2.** Thank you for point out that our previous proofs are difficult to comprehend. To address this issue, we now adopt a classical perspective of low-rank approximation to prove that class embeddings are transformed to be the principal directions by the attention operations. Please refer to the Interpretation Outline section of our Common Response for our rephrased thoughts and conclusions. In short, we prove that the principal directions are a good low-rank approximation to the patch embeddings in terms of coding rate. Therefore, we replace equation (20) with $\min_{\boldsymbol{Q}} |R(\boldsymbol{Z})-R(\boldsymbol{Q})|$ $\text{s.t}.$ $\text{rank}(\boldsymbol{Q})=C$, which is a more intuitive objective motivated by the idea of low-rank approximation. By the geometric interpretation of the coding rate, we will have that $R(\boldsymbol{Z}) \geq R(\boldsymbol{Q})$, constraining that each class embeddings lie in the subspace of patch embeddings. Therefore, the objective becomes $\max_{\boldsymbol{Q}} R(\boldsymbol{Q})$, and it derives a self-attention-like operation on $\boldsymbol{Q}$. For keeping the constraint and building up the connection between $\boldsymbol{Q}$ and $\boldsymbol{Z}$, we can replace the first and the second $\boldsymbol{Q}$ in the self-attention $\boldsymbol{Q}(\boldsymbol{Q}^T \boldsymbol{Q})$ with $\boldsymbol{Z}$ and get $\boldsymbol{Z}(\boldsymbol{Z}^T \boldsymbol{Q})$, a cross-attention-like operation. Notice that the output of such an operation always lies in the subspace of $\boldsymbol{Z}$. It is easy to prove that the scaled leading $C$ principal directions are an optimal solution for the above low-rank approximation objective. Although we believe that current line of thought is intuitive, we are willing to provide detailed proofs if reviewers request.
>
> [1] Learning Diverse and Discriminative Representations via the Principle of Maximal Coding Rate Reduction. Yaodong Yu, Kwan Ho Ryan Chan, Chong You, Chaobing Song, Yi Ma. NeurIPS 2020.

---

> > ### Comment · Reviewer_V3V2 · 2024-08-13
> >
> > Thank you for the detailed response, which substantially addresses the theoretical aspects of my concerns. Consequently, I am inclined to adjust my review favorably regarding these points. However, given the limited experimental validation, I recommend that more comprehensive experiments be included in the final version.

---

> > > ### Author Response · Authors · 2024-08-14
> > >
> > > Thank you for your responses and acknowledgment on our rebuttal and greatly appreciate your increased rating. We have reported new experimental progress in the official comment to the Author Rebuttal. We wish these experiments would address your concerns on experimental evaluation. We will definitely continue to enhance the experimental section by including these new results in the final version. Thanks a lot for your valuable suggestions, which have helped us improve the quality of our research.

---

> ### Comment · Area_Chair_GsUq · 2024-08-13
>
> Dear Reviewer V3V2
>
> Thanks for reviewing this work. Would you mind to check authors' feedback and see if it resolves your concerns or you may have further comments?
>
> Best wishes
>
> AC

---

### Official Review · Reviewer_MHpG · 2024-07-11

**Soundness:** 2
**Presentation:** 2
**Contribution:** 3
**Rating:** 5
**Confidence:** 4

**Summary:**

This paper derives white-box decoders for Transformer-based semantic segmentation from a compression perspective, which links the process to Principal Component Analysis (PCA). The authors introduce DEPICT, which clarifies the mechanisms of black-box decoders and achieves comparable performance with significantly fewer parameters. Using ViT-B as the encoder, DEPICT outperforms traditional mask transformers on the ADE20K dataset.

**Strengths:**

The paper provides a solid theoretical foundation by analyzing mathematically complex decoders used in Transformer-based semantic segmentation.

This method bridges the gap between theoretical understanding and practical application.

In addition, DEPICT decoders achieve comparable performance to traditional black-box decoders while using significantly fewer parameters.

**Weaknesses:**

The primary drawback of this paper is the lack of extensive experimentation with existing decoder architectures. While the focus on theoretical derivation is understandable, the current experimental results are insufficient to confirm that the proposed DEPICT algorithm operates as claimed by the authors. A more comprehensive comparison across various datasets and architectures is necessary to validate the effectiveness and generalizability of DEPICT.

**Questions:**

1.	Please include the Cityscapes dataset, which is widely used in semantic segmentation, in your experiments.
2.	Could you compare the performance of the proposed method with the following prominent Transformer decoder-based methods to see if your approach can be applied and how it performs against them?
	1)	SegFormer (Xie et al. 2021)
	2)	Segmenter (Strudel et al. 2021)
	3)	Mask2Former (Cheng et al. 2022)
	4)	FeedFormer (Shim et al. 2023)
3.	In Transformer decoder structures, GFLOPs (floating-point operations per second) are more crucial than the number of parameters due to the attention mechanism. Please include GFLOPs in your comparisons.

If these points are addressed during the revision period, I would be willing to consider increasing my evaluation score.

**Limitations:**

The most significant limitation of this paper is the lack of extensive experiments to validate the performance of the proposed algorithm. Additionally, the paper does not consider the FLOPs metric, which is crucial for evaluating efficiency.
Please address the points mentioned in the questions section to strengthen the validation of your proposed DEPICT algorithm.

---

> ### Author Rebuttal · Authors · 2024-08-07
>
> Thank you for your valuable suggestions. As the reviewer suggested, we have further validated our proposed interpretation and evaluated the effectiveness of our DEPICT on a set of new experiments on datasets Cityscapes and Pascal Context, including ablating the variants, testing the robustness under parameter perturbation and evaluating GFLOPs. We report the performance of our modifications to DEPICT, their justifications, detailed experimental results and an outline of our interpretation in our Common Response.
>
> Here, we address the concerns listed by reviewer MHpG.
>
> - First of all, we evaluate the GFLOPs, the amount of floating point operations, of only the decoders for fair comparison. Compared with Segmenter's decoder (Mask Transformer), the GFLOPs of our DEPICT is 2/3/4/10 versus Mask Transformer's 2/6/22/37 when both based on ViT-Tiny/Small/Base/Large. Compared to MaskFormer's decoder, the GFLOPs of our DEPICT is 10 based on ViT-Large; whereas it is at least 270 for MaskFormer based on Swin-Large (total: 375 GFLOPs [1] - Swin-L: more than 104 GFLOPs [2]). These evidences demonstrate that our DEPICT indeed serves an extremely light-weight decoder with comparable even better mIoU. Specifically, we must express our apologies for a vital typo that we mis-spelled "Mask Transformer" as "MaskFormer" in our submitted paper. Considering that, "Mask Transformer" has been widely used in various papers, we would use "Segmenter's decoder" instead in the final version of our paper.
>
> - Then, we report our newly added experimental results on Cityscapes and Pascal Context.  Due to limited time during rebuttal phase, we did not report the results of ViT-Large based DEPICT on these two datasets, but we will get it done soon (during the discussion period or at least in the final version). Based on ViT-Small, we can observe that our DEPICT outperforms Segmenter by 7% mIoU on both Cityscapes and Pascal Context.
>
> - Third, as suggested by reviewers, we would like to provide a set of direct comparisons between DEPICT and more advanced black-box methods. Nevertheless, we  argue that such kind of comparisons currently are unfair for our DEPICT. For example, on ADE20K, MaskFormer's Swin-L based decoder achieves 54.1% mIoU with at least 270 GFLOPs; whereas our ViT-L (lagging behind Swin-L by 2% top1 acc) based DEPICT achieves 52.5% mIoU with merely 10 GFLOPs. Meanwhile, MaskFormer uses a linear combination of the focal loss and the dice loss; whereas our DEPICT merely uses the standard CE loss without weight rebalancing. If there is no time limitation, we would like to attempt to address all these issues for fair comparisons. In addition,
> we now report better performance of our newly modified DEPICT, compared to Segmenter (the only fair comparison). Details can be found in our Common Response.
>
> - For applying our interpretation to more black-box methods, it seems a potential but challenging future work. The most thorny part is prevalent methods typically involving a pixel decoder (see Fig.3 of [3]), resulting in extremely more GFLOPs and parameters, and remaining uninterpretable from our proposed perspective. Our work implies that there must exist more efficient methods without the burdensome pixel decoder.
>
> Most importantly, as suggested by the reviewer, we add several interesting experiments to confirm that our DEPICT indeed performs as we claimed. Here, we would like to list all current evidences.
>
> + We strictly implement the *MSSA* block (see Line 171 for previous implementation) and remove the *ISTA* block (see Line 197 for context), enabling us to conclude that all DEPICT does is maximizing the projected coding rate on subspaces spanned by leading principal directions or minimizing it on subspaces spanned by remaining principal directions. By the sign of the learned step-size parameter, it is a white box for us to find out whether a specific block is maximizing it or minimizing it.
>
> + According to above point, all the learned step size should be positive if we let all blocks model the ambient space rather than low-dimensional subspaces. We conduct a set of experiments in this extreme case by setting $\text{heads}=1$ and $\text{dim}$_$\text{head}=\text{dim}$, and the result is as expected.
> + We visualize all the class embeddings output by a cross-attention block across images and find that principal directions lies in a union of $C$ orthogonal subspaces.
> + During the inference phase, we generate the $\text{heads}$ random orthogonal matrices per image and use them to transform the parameter matrices of each head individually. We observe that DEPICT based on various ViT backbones shows no accuracy drop under such a perturbation. Furthermore, we generate one random orthogonal matrix to transform the parameter matrix of the entire block. Despite parameters are perturbed across heads, DEPICT based on various ViT backbones still shows limited drop in mIoU (less than 3% for ViT-L based DEPICT). Such robustness strongly validate the interpretation that our attention blocks essentially model subspaces thus performing orthogonal transformations on their learned bases will not lead to collapsed performance.
>
> We hope  the clarifications above could resolve your concerns on our work. Please let us know if any further clarification is needed.
>
>
> [1] Per-Pixel Classification is Not All You Need for Semantic Segmentation. Bowen Cheng, Alex Schwing, Alexander Kirillov. NeurIPS 2021.
>
> [2] Swin Transformer: Hierarchical Vision Transformer using Shifted Windows. Ze Liu, Yutong Lin, Yue Cao,  Han Hu, Yixuan Wei, Zheng Zhang, Stephen Lin, Baining Guo. ICCV 2021.
>
> [3] Transformer-Based Visual Segmentation: A Survey. Xiangtai Li, Henghui Ding, HaoboYuan, Wenwei Zhang, Jiangmiao Pang,  Guangliang Cheng, Kai Chen, Ziwei Liu, Chen Change Loy. TPAMI 2024.
>
> [4] White-box transformers via sparse rate reduction. Yaodong Yu, Sam Buchanan, Druv Pai, Tianzhe Chu, Ziyang Wu, Shengbang Tong, Benjamin Haeffele, Yi Ma. NeurIP 2023.

---

> > ### Comment · Reviewer_MHpG · 2024-08-13
> > **Post-rebuttal comments**
> >
> > The authors have conducted many of the experiments I suggested, but it’s unfortunate that they were unable to perform additional experiments on various models and datasets like Cityscapes due to time constraints. I am raising my score to borderline accept, but if the paper is rejected, I strongly recommend including these experiments in detail for the next submission.

---

> > > ### Author Response · Authors · 2024-08-14
> > >
> > > Thank you for your responses and acknowledgment on our rebuttal, and highly appreciate your increased rating. We have reported new experimental progress in the official comment to the Author Rebuttal. While we believe that these additional experiments have strengthened our work, we will continue to conduct further experiments as suggested. We sincerely appreciate your valuable feedbacks, which have helped us improve our work.

---

> ### Comment · Area_Chair_GsUq · 2024-08-13
>
> Dear Reviewer MHpG
>
> Thanks for reviewing this work. Would you mind to check authors' feedback and see if it resolves your concerns or you may have further comments?
>
> Best wishes
>
> AC

---

### Official Review · Reviewer_ZoMh · 2024-07-16

**Soundness:** 3
**Presentation:** 3
**Contribution:** 2
**Rating:** 5
**Confidence:** 3

**Summary:**

This work attempts to view Transformer-based semantic segmentation from the perspective of principal component analysis and develops interpretable Transformer decoders for segmentation based on this insight, which can achieve comparable performance to their black-box counterparts with significantly fewer parameters.

**Strengths:**

1. The presentation is overall clear and the derivation is solid;
2. It is impressive to achieve comparable performance with fewer parameters and interpretability;
3. The visualization of learned parameters in derived architecture validates the assumption and derivation.

**Weaknesses:**

1. The authors seem to lack awareness of recent advances in segmentation and have significant misunderstanding about Transformer-based decoders for segmentation. L34-35 says "these decoders typically set a learnable class embedding for each predefined class." However, in the four citations at L78 ([27,6,5,13]), only the first paper adopts this paradigm. The other works do not include learnable class embeddings for predefined classes but use “zero-initialized query features with positional embeddings corresponding to instances/segments, instead of classes. Thus, the formulation in this paper does not apply to any of them. Additionally, it is incorrectly stated in L246 that MaskFormer serves as the decoder for Segmenter.
2. The comparison in Figure 6 does not use the most advanced methods as the baseline, making the results less convincing. Besides, the linear decoder achieves performance comparable to the black-box counterpart with only about one percent of the parameters. This somewhat implies that the results presented in this work are trivial.
3. The authors claim that there is principled guidance to set hyper-parameters such as num_heads and dim_heads. However, does this constraint ensure consistently satisfactory performance without the need for laborious manual tuning? Or is it merely to reduce the hyper-parameter tuning space to align with theoretical derivations? If it is the former, it would be beneficial to report results on a broader range of datasets to demonstrate its generality.

**Questions:**

See weakness.

**Limitations:**

Although the paper presents a very fascinating theoretical framework, the authors' ignorance and misunderstanding of the basic Transformer-based segmentation models make many claims in this paper is incorrect. And the formulation of the baseline black-box this manuscript focus on is not sufficiently representative.

---

> ### Author Rebuttal · Authors · 2024-08-07
>
> Thank you for your valuable comments. It is indeed that the meta-architecture we investigated sounds not representative enough for a vast range of Transformer-based decoders. However, the insights of our work is capable to be generalized to partly interpret them.
>
> First of all, we express our sincere appreciation for pointing out our mistakes. We will adopt the meta architecture proposed by [1] (see its Fig.3) as the truly representative one and admit that our interpretation is investigating a simplified one in the paper. Additionally, we will replace "MaskFormer" with "Segmenter" in the table. The decoder of Segmenter is called "Mask Transformer" but we mistakenly spelled it. Considering that, "Mask Transformer" has been widely used in various paper, we would not to use it in the final version of the paper in order to avoid naming ambiguity.
>
> Now, we focus on reporting how we attempt to address the concerns listed by reviewer.
>
> - We would like to show that addressing a limitation of our current interpretation will result in replacing class embeddings with segment/instance embeddings. That is, using class embeddings couples each class with a principal direction, such that each class strictly corresponds to one principal direction. However, an image typically contains a small part of all classes and each class allows rich intra-class variance thus demands more than one principal directions to representing itself. Therefore, it is more reasonable to introduce segment or instance as a finer-grained and more flexible concept to replace the current role of class.  As evidence, we find that there are still higher correlations among class embeddings output by our DEPICT. Although  such correlations can be significantly eased by using cross-attention, it leads to a drop in mIoU. Additionally,  Mask2Former reports that making query features learnable raises mIoU by 1.8% (see its Table 4. (b)). From the perspective of our work, it is because that the attention block performs a gradient descent step and a good initialization to start would be beneficial.
>
> - As suggested by reviewers, we would like to provide a set of direct comparisons between DEPICT and more advanced black-box methods. Nevertheless, we have to argue that such kind of comparisons currently are unfair for our DEPICT. For example, on ADE20K, MaskFormer's Swin-L based decoder achieves 54.1% mIoU with at least 270 GFLOPs; whereas our ViT-L (lagging behind Swin-L by 2% top1 acc) based DEPICT achieves 52.5% mIoU with merely 10 GFLOPs. Meanwhile, MaskFormer uses a linear combination of the focal loss and the dice loss; whereas our DEPICT merely uses the standard CE loss without weight rebalancing. If there is no time limitation, we would like to attempt to address all these issues for fair comparisons. As for the linear decoder, it is not scalable not only on layer backbone but also to trade the amount of parameters for better mIoU. For example, it lags behind our DEPICT by 1.8% mIoU when based on ViT-Large and simply adding more linear layers does not change the performance. (These results are based on our newly modified DEPICT, and have reported in our Common Response.)
>
> - We actually find that adding constraint $\text{heads}*\text{dim}$_$\text{head}=C$ is unreasonable, as its derivation considered only one image whereas there are various principal directions across images of the whole dataset. Therefore, we have removed this constraint. Then, we conduct experiments to evaluate the impact of using a ranging number of heads, and find that using a relatively small number of heads (i.e., it is 3 for ADE20K) leads to the best performance, implying that a tighter lower bound leads to less effective training.
>
> - We report more experimental results on Cityscapes and Pascal Context. Limited by time, We did not report the result of ViT-Large based DEPICT on these two datasets, but we will get it done soon (during the discussion phase or at least in the final version). Our DEPICT based on ViT-Small can outperform Segmenter by 7% mIoU on both Cityscapes and Pascal Context.
>
> We hope that our clarification points above could resolve your concerns on our work. Please let us know if any further clarification is needed.
>
> [1] Transformer-Based Visual Segmentation: A Survey. Xiangtai Li, Henghui Ding, HaoboYuan, Wenwei Zhang, Jiangmiao Pang,  Guangliang Cheng, Kai Chen, Ziwei Liu, Chen Change Loy. TPAMI 2024.

---

> > ### Comment · Reviewer_ZoMh · 2024-08-13
> >
> > The author’s response solves most of my questions and concerns. Therefore, I would like to keep my original positive score.

---

> > > ### Author Response · Authors · 2024-08-14
> > >
> > > Thank you for your responses and acknowledgment on our rebuttal. Your valuable feedbacks on our work have undoubtedly helped us a lot.

---

> ### Comment · Area_Chair_GsUq · 2024-08-13
>
> Dear Reviewer ZoMh
>
> Thanks for reviewing this work. Would you mind to check authors' feedback and see if it resolves your concerns or you may have further comments?
>
> Best wishes
>
> AC

---

### Official Review · Reviewer_CNaF · 2024-07-17

**Soundness:** 3
**Presentation:** 3
**Contribution:** 3
**Rating:** 6
**Confidence:** 4

**Summary:**

A view of compression from feature space with dimension $m$ to category $c$ is proposed for semantic segmentation. With this formulation of a PCA-like operation, a white-box encoder is designed with self-attention or cross-attention. Experiments show that the proposed encoder with ViT gets better performance than Maskformer.

**Strengths:**

1. The view of compression for semantic segmentation is interesting and it is verified by visualization of the classifier $P$.
2. Experiments verify the effectiveness of the encoder.

**Weaknesses:**

1. Compression is not the original idea of this paper and it is inspired by [37], so I do not give a very high review score. However, I still think this work is good.
2. It's better to give results on another dataset to show the universal ability.
3. It lacks comparison with the new SOTA semantic segmentation approaches. Only the maskformer is compared.
4. "A s" in Line 241 --> "As".

**Questions:**

1. Is it possible to adapt the PCA analysis to the CNN-based model?
2. The number of hyperparameters of the transformer block is very few. If the dim_head and heads are set to 1 and $c$ in the white-box encoder, will it limit the diversity of the model?

---

> ### Author Rebuttal · Authors · 2024-08-07
>
> Thank you for your appreciation of our work. As suggested, we add experiments on more datasets, including Cityscapes and Pascal Context, and find that our DEPICT performs consistently better that of its black-box counterparts, i.e., the decoder of Segmenter. And we would like to compare our DEPICT to more advanced methods, showing that our DEPICT serves as an extremely light-weight decoder with acceptable lower mIoU. Specifically, we express our apologies for a typo that mis-spelled "Mask Transformer" as "MaskFormer" in the paper. Since that "Mask Transformer" has been widely used in various papers, we would use "Segmenter's decoder" instead in the final version of our paper.
>
> As reported in details in our Common Responses, we have provided evaluations on more datasets to improve the empirical evaluations as Reviewer CNaF suggested. To make a fair comparison, we strictly implemented the *MSSA* block (see Line 171 for previous implementation) and removed the *ISTA* block (see Line 197 for context). These experimental results show that our DEPICT now consistently outperforms Segmenter's decoder with further reduced amount of parameters. While the white-box nature of our DEPICT has been strengthened by the above-mentioned modifications, we further conduct more experiments to validate our interpretation.
>
> Here, we focus on reporting how we have addressed the concerns listed in the review comments.
>
> As raised by reviewer, the idea of quantifying compression by coding rate is indeed not the novelty of our work and, we did not claim that as one of our novel contributions. To make our novel contributions more clear, we would like to make the following interpretation (or clarification). Compared to [1], our novelty and distinct contributions are:
> + We introduce the idea of compression to design an interpretable semantic segmentation and propose an interpretation framework from the perspective of the classical *PCA*.
> + For the first time, we derive the cross-attention variant of the MSSA block and we conduct experiments to evaluate the performance of its strict implementation, showing improved performance (also see B.1.1 of [1] for its relaxed implementation).
>
> Second, we would like to report more experimental results on datasets Cityscapes and Pascal Context.  Due to limited time during rebuttal phase, we do not report the results of ViT-Large based DEPICT on these two datasets at this moment, but we will get it done soon (during the discussion, or at least in the final version). We report the results based on ViT-Small, and show that our DEPICT again outperforms Segmenter by 7% mIoU on both Cityscapes and Pascal Context.
>
> Third, as suggested by reviewers, we would like to provide a set of direct comparisons between DEPICT and more advanced black-box methods. Nevertheless, we have to argue that such kind of comparisons currently are unfair for our DEPICT. For example, on ADE20K, MaskFormer's Swin-L based decoder achieves 54.1\% mIoU with at least 270 GFLOPs; while our ViT-L (lagging behind Swin-L by 2% top1 acc) based DEPICT achieves 52.5\% mIoU with merely 10 GFLOPs. Meanwhile, MaskFormer uses a linear combination of the focal loss and the dice loss; whereas our DEPICT merely uses the standard CE loss without weight rebalancing. If there is no time limitation, we would like to attempt to address all these issues for fair comparisons.
>
> - For Question #1: "Is it possible to adapt PCA to CNN-based models?" This is an insightful but challenging question. It seems to us that the answer is yes. According to [2], a self-attention layer can express any convolutional layer. Conversely, a self-attention operation may be decomposed into convolutional filters. Additionally, in fully convolutional networks, classification presents as features output by the last layer. In other words, there seems no difference between extracting features and giving classification, for that is right the reason allowing us to introduce PCA for interpretation.
>
> - For Question #2: we actually find that adding constraint $\text{heads}*\text{dim}$_$\text{head}=C$ is unreasonable, as its derivation considered only one image whereas there are various principal directions across images of the whole dataset. Therefore, we removed this constraint. Then, we conduct experiments to evaluate the impact of using a ranging number of heads, and find that a relatively small number of heads (i.e., 3 for ADE20K) renders the best performance, implying that a tighter lower bound leads to less effective training. Using a large number of heads not only causes a heavier occupation on GPU memory but also results in inferior performance. Please let us know if we misunderstand your question. Thanks so much!
>
> We hope that our point-by-point responses above have resolved your concerns on our work and, highly appreciate your encouragement in the comments and the pre-rating. Please let us know if any further clarification is needed.
>
> [1] White-box transformers via sparse rate reduction. Yaodong Yu, Sam Buchanan, Druv Pai, Tianzhe Chu, Ziyang Wu, Shengbang Tong, Benjamin Haeffele, Yi Ma. NeurIP 2023.
>
> [2] On the relationship between self-attention and convolutional layers. Jean-Baptiste Cordonnier, Andreas Loukas, Martin Jaggi. ICLR 2020.

---

> ### Comment · Area_Chair_GsUq · 2024-08-13
>
> Dear Reviewer CNaF
>
> Thanks for reviewing this work. Would you mind to check authors' feedback and see if it resolves your concerns or you may have further comments?
>
> Best wishes
>
> AC

---

### Author Rebuttal · Authors · 2024-08-07

We thank all reviewers for the constructive suggestions and insightful comments. We appreciate the weaknesses and limitations spotted by reviewers. To address these issues, we would like to provide more theoretical and experimental results. Briefly, we further solidify our compression-based interpretation by presenting more rigorous proofs, and achieve much better experimental results while strictly implementing the derivations. As requested by reviewers, we conduct new experiments to further validate the effectiveness and mechanism of our proposed white-box decoders.

+ For compression with less information loss, we maximize *the projected coding rate* respectively on $C$ principal directions. As it is costly to directly maximize them, we assumed that the projected coding rate on a subspace spanned by them serves as the lower bound of their sum. Now, we rigorously prove this point holds.

+ Aligning with [1], the utilized *MSSA* block involved an independently learned parameter matrix for implementation simplicity (see B.1.1 in [1]). We now remove it to guarantee the model functions as our interpretation suggests. Meanwhile, as the *ISTA* block lacked convincing justification for its utilization, it is also removed, meaning that DEPICT is now fully attention-based. While strengthening the interpretability of our proposed DEPICT, these two modifications also significantly improve the segmentation performance and reduce the parameter amount.

+ By the decoder, class embeddings representing the same class vary across images. Such variability is desirable for allowing rich intra-class variation and partly explain why query decoder is superior to linear decoder. Back to our work, we constrained $\text{heads}* \text{dim}$\_$\text{head}=C$ as there are at most $C$ classes and each of them is represented by one principal direction. However, it merely holds for a single image. For the whole dataset, class embeddings lie in a union of $C$ orthogonal subspaces and each subspace is spanned by the class embeddings representing a specific class. Therefore, we remove the constraint that $\text{heads}* \text{dim}$\_$\text{head}=C$.

+ We now integrate the proof for solving principal directions into the coding rate framework by providing a more generalized and insightful proof from the perspective of *low-rank approximation*. In short, the scaled principal directions approximate patch embeddings well in terms of coding rate.

# Key Contributions
Our central contribution is that we introduce a compression-based interpretation framework for Transformer-based semantic segmentation decoders. Although the proofs are complicated, the conclusions are simple and intuitive: self-attention prepares patch embeddings for compression and cross-attention seeks principal directions as classifiers. Based on such insights, we design white-box decoders, DEPICT, which remarkably outperform their black-box counterparts in terms of mIoU, #params, and GFLOPS. In summary, our work serves as a first yet solid step toward developing a comprehensive interpretation framework for semantic segmentation. We believe that our insights and findings are worth sharing with the community to encourage further efforts toward better white-box decoders.

# New Experiments
Here, we provide a brief summary of the new experiments and their results for the reviewers. We compare our modified DEPICT with Segmenter on ADE20K, Cityscapes, and Pascal Context, and find that DEPICT consistently performs better. We test the robustness of DEPICT under parameter perturbation, and observe that DEPICT shows very limited drop in mIoU. We ablate variants of DEPICT and find that the currently adopted fully attention-based design performs the best. Please refer to the attached .pdf for detailed illustrations and results. We are willing to provide more detailed experimental results if requested by the reviewers.

[1] White-box transformers via sparse rate reduction. Yaodong Yu, Sam Buchanan, Druv Pai, Tianzhe Chu, Ziyang Wu, Shengbang Tong, Benjamin Haeffele, Yi Ma. NeurIP 2023.

---

> ### Author Response · Authors · 2024-08-14
>
> We highly appreciate the reviewers' acknowledgment on our rebuttal as well as the efforts of AC. We are also appreciate the feedbacks from all reviewers, which help us to the quality of our work. In particular, we appreciate the suggestions regarding the experiments. Since the beginning of the rebuttal and discussion period, we started to conduct additional experiments and now all the aforementioned experiments are completed. To be specific:
>
> + We measured the coding rate of subspaces modeled by the decoder across layers and found that the coding rate gradually concentrates on these modeled subspaces, validating that the decoder operates as our interpretation suggests.
>
> + We tried to segment the image using the exact principal directions rather than the class embeddings calculated by the decoder. Surprisingly, we found that the principal directions serve as excellent classifiers, as their segmentation clearly highlights objects that the trained models failed to detect.
>
> + We completed the required experiments on the Cityscapes and Pascal Context datasets, and are quite pleased to report that DEPICT remains strongly comparable to, and in some cases even surpasses, Segmenter on these two datasets.
>
> In the final version of the paper, we will include all the experiments reported during the rebuttal and discussion period.
>
> Below, we present the comparative results of DEPICT and Segmenter across three datasets. Note that although we did not have time to fine-tune the hyperparameters of DEPICT on the latter two datasets, DEPICT remains strongly comparable to, and even exceeds, Segmenter. In particular, on Pascal Context, DEPICT based on ViT-B outperforms Segmenter based on DeiT-B.
> | Model&backbone      |ADE20K mIoU (ss/ms)         | Cityscape mIoU (ss/ms) | Pascal Context mIoU (ss/ms)|
> | -------- | -------- | ---------- |-------|
> | **DEPICT-ViT-T**    |**39.3/40.7**|  **73.5/-** |  46.9/-         |
> | **DEPICT-ViT-S**    |**46.7/47.7**|  **76.8/-** |  **52.8/53.4**      |
> | **DEPICT-ViT-B**    |**49.2/50.7**|  77.6/-     |  **54.8/55.3**     |
> | **DEPICT-DeiT-B**   |-/-|  78.6/80.5     |   54.4/54.8         |
> | **DEPICT-ViT-L**    |**52.5/54.0**| 78.9/- |   57.3/58.3        |
> | **Segmenter-ViT-T** |38.1/38.8|72.7/75.0|  47.1/-     |
> | **Segmenter-ViT-S** |45.3/46.9|  76.4/78.9    |    52.1/-       |
> | **Segmenter-ViT-B** |48.5/50.0| -/-   |    -/-     |
> | **Segmenter-DeiT-B** |48.7/50.1| 78.7/80.6    |    -/55.0        |
> | **Segmenter-ViT-L** |51.8/53.6|  79.1/81.2     |   58.1/59.0         |

---

### Decision · Program_Chairs · 2024-09-25

**Decision:**

Accept (poster)

**Comment:**

This paper derives white-box decoders for Transformer-based semantic segmentation from the perspective of compression.
Initially it received rating of 6544.
Concerns are about theoretical aspects and more experiment validation.
Most of the concerns were resolved during rebuttal and the reviewers raised the score to 6555.
The AC recommends acceptance for this paper.
The authors are urged to include more comprehensive experiments (eg, Cityscapes) as requested by reviewer V3V2 and reviewer MHpG in the camera-ready version of the paper.